# ON THE NEURAL TANGENT KERNEL OF EQUILIBRIUM MODELS

## ABSTRACT

This work studies the neural tangent kernel (NTK) of the deep equilibrium (DEQ) model, a practical "infinite-depth" architecture which directly computes the infinite-depth limit of a weight-tied network via root-finding. Even though the NTK of a fully-connected neural network can be stochastic if its width and depth both tend to infinity simultaneously, we show that contrarily a DEQ model still enjoys a deterministic NTK despite its width and depth going to infinity at the same time under mild conditions. Moreover, this deterministic NTK can be found efficiently via root-finding.

## 1 INTRODUCTION

Implicit models form a new class of machine learning models where instead of stacking explicit "layers", they output $z$ s.t $g(x, z) = 0$, where $g$ can be either a fixed point equation (Bai et al., 2019), a differential equation (Chen et al., 2018b), or an optimization problem (Gould et al., 2019). This work focuses on deep equilibrium models, a class of models that effectively represent a "infinite-depth" weight-tied network with input injection. Specifically, let $f_\theta$ be a network parameterized by $\theta$, let $x$ be an input injection, DEQ finds $z^*$ such that $f(z^*, x) = z^*$, and uses $z^*$ as the input for downstream tasks. One interesting question to ask is, what will DEQs become if their widths also go to infinity? It is well-known that at certain random initialization, neural networks of various structures converge to Gaussian processes as their widths go to infinity (Neal, 1996; Lee et al., 2017; Yang, 2019; Matthews et al., 2018; Novak et al., 2018; Garriga-Alonso et al., 2018). Recent deep learning theory advances have also shown that in the infinite width limit, with proper initialization (the NTK initialization), training the network $f_\theta$ with gradient descent is equivalent to solving kernel regression with respect to the neural tangent kernel (NTK) (Arora et al., 2019; Jacot et al., 2018; Yang, 2019; Huang et al., 2020). These kernel regimes provide important insights to understanding how neural networks work.

However, the infinite depth (denote depth as $d$) regime introduces several caveats. Since the NTK correlates with the infinite width (denote width as $n$) limit, a question naturally arises as how do we let $n, d \to \infty$? Hanin & Nica (2019) proved that as long as $d/n \in (0, \infty)$, the NTK of vanilla fully-connected neural network (FCNN) becomes stochastic. On the other hand, if we first take the $n \to \infty$, then $d \to \infty$[1], Jacot et al. (2019) showed that the NTK of a FCNN converges either to a constant (*freeze*), or to the Kronecker Delta (*chaos*). In this work, we prove that with proper initialization, the NTK-of-DEQ enjoys a limit exchanging property $\lim_{d \to \infty} \lim_{n \to \infty} \Theta_n^{(d)}(x, y) = \lim_{n \to \infty} \lim_{d \to \infty} \Theta_n^{(d)}(x, y)$ with high probability, where $\Theta_n^{(d)}$ denotes the empirical NTK of a neural network with $d$ layers and $n$ neurons each layer. Intuitively, we name the left hand side "DEQ-of-NTK" and the right hand side "NTK-of-DEQ". The NTK-of-DEQ converges to meaningful deterministic fixed points that depend on the input in a non-trivial way, thus avoiding the freeze vs. chaos scenario. Furthermore, analogous to DEQ models, we can compute these kernels by solving fixed point equations, rather than iteratively applying the updates as for traditional NTK. We evaluate our approach and demonstrate that it matches the performance of existing regularized NTK methods.

---

[1] The computed quantity is $\lim_{d \to \infty} \lim_{n \to \infty} \Theta_n^{(d)}(x, y)$.

## 2 BACKGROUND AND PRELIMINARIES

A vanilla FCNN has the form $g^{(t)} = \sigma(W^{(t)}g^{(t-1)} + b^{(t)})$ for the $t$-th layer, and in principle $t$ can be as large as one wants. A weight-tied FCNN with input injection (FCNN-IJ) makes the bias term related to the original input and ties the weight in each layer by taking the form $z^{(t)} := f(z^{(t-1)}, x) = \sigma(Wz^{(t-1)} + Ux + b)$. Bai et al. (2019) proposed the DEQ model, which can be equivalent to running an infinite-depth FCNN-IJ, but updated in a more clever way. The forward pass of DEQ is done by solving $f(z^*, x) = z^*$. For a stable system, this is equivalent to solving $\lim_{t\to\infty} f^{(t)}(z^{(0)}, x)$. The backward iteration is done by computing $df(z^*, x)/dz^*$ directly through the implicit function theorem, thus avoiding storing the Jacobian for each layer. This method traces back to some of the original work in recurrent backpropagation (Almeida, 1990; Pineda, 1988), but with specific emphasis on: 1) computing the fixed point directly via root-finding rather than forward iteration; and 2) incorporating the elements from modern deep networks in the single "layer", such as self-attention transformers (Bai et al., 2019), multi-scale convolutions (Bai et al., 2020), etc. DEQ models achieve nearly state-of-the-art performances on many large-scale tasks including the CityScape semantic segmentation and ImageNet classification, while only requiring constant memory. Although a general DEQ model does not always guarantee to find a stable fixed point, with careful parameterization and update method, monotone operator DEQs can ensure the existence of a unique stable fixed point (Winston & Kolter, 2020).

The study of large width limits of neural networks dates back to Neal (1996), who first discovered that a single-layered network with randomly initialized parameters becomes a Gaussian process (GP) in the large width limit. Such connection between neural networks and GP was later extended to multiple layers (Lee et al., 2017; Matthews et al., 2018) and various other architectures (Yang, 2019; Novak et al., 2018; Garriga-Alonso et al., 2018). The networks studied in this line of works are randomly initialized, and the GP kernels they induce are often referred to as the NNGP.

A line of closely-related yet orthogonal work to ours is the mean-field theory of neural networks. This line of work studies the relation between depth and large-width networks (hence a GP kernel in limit) at initialization. Poole et al. (2016); Schoenholz et al. (2016) showed that at initialization, the correlations between all inputs on an infinitely wide network become either perfectly correlated (*order*) or decorrelated (*chaos*) as depth increases. They suggested we should initialize the neural network on the "edge-of-chaos" to make sure that signals can propagate deep enough in the forward direction, and the gradient does not vanish or explode during backpropagation (Raghu et al., 2017; Schoenholz et al., 2016). These mean-field behaviors were later proven for various other structures like RNNs, CNNs, and NTKs as well (Chen et al., 2018a; Xiao et al., 2018; Gilboa et al., 2019; Hayou et al., 2019). We emphasize that despite the similar appearance, our setting avoids the order vs. chaos scheme completely by adding input injection. The injection guarantees the converged NTK depends nontrivially on the inputs, as we will see later in the experiments.

While previous results hold either only at initialization or networks with only last layer trained, analogous limiting behavior was proven by Jacot et al. (2018) to hold for fully-trained networks as well. They showed the kernel induced by a fully-trained infinite-width network is the following:

$$\Theta(x, y) = \mathbb{E}_{\theta \sim \mathcal{N}} \left[ \left\langle \frac{\partial f(\theta, x)}{\partial \theta}, \frac{\partial f(\theta, y)}{\partial \theta} \right\rangle \right], \tag{1}$$

where $\mathcal{N}$ represents the Gaussian distribution. They also gave a recursive formulation for the NTK of FCNN. Arora et al. (2019); Alemohammad et al. (2020); Yang (2020) later provided formulation for convolutional NTK, recurrent NTK, and other structures.

One may ask what happens if both the width and the depth in a fully-trained network go to infinity. This question requires careful formulations as one should consider the order of two limits, as Hanin & Nica (2019) proved that width and depth cannot simultaneously tend to infinity and result in a deterministic NTK, suggesting one cannot always swap the two limits. An interesting example is that Huang et al. (2020) showed that the infinite depth limit of a ResNet-NTK is deterministic, but if we let the width and depth go to infinity at the same rate, the ResNet behaves in a log-Gaussian fashion (Li et al., 2021). Meanwhile, the infinite depth limit of NTK does not always present favorable properties. It turns out that the vanilla FCNN does not have a meaningful convergence: either it gives a constant kernel or the Kronecker Delta kernel (Jacot et al., 2019).

**Our contributions.** We first show that unlike the infinite depth limit of NTK to FCNN, the *DEQ-of-NTK* does not converge to a degenerate kernel. This non-trivial kernel can be computed efficiently using root-finding. Moreover, the *NTK-of-DEQ* coincides with the DEQ-of-NTK under mild conditions. Although the proofs here involved infinite limits, we also show numerically that reasonably large networks converge to roughly the same quantities as predicted by theory, and we show the NTK-of-DEQ matches the performances of other NTKs on real-world datasets.

## 2.1 NOTATION

We write capital letter $W$ to represent matrices or tensors, which should be clear from the context, and use $[W]_i$ to represent the element of $W$ indexed by $i$. We write lower case letter $x$ to represent vectors or scalars. For $a \in \mathbb{Z}_+$, let $[a] = \{1, \ldots, a\}$. Denote $\sigma(x) = \sqrt{2}\max(0, x)$ as the normalized ReLU and $\dot\sigma$ its derivative (which only needs to be well-defined almost everywhere). The symbol $\sigma_a^2$ with subscript is always used to denote the variance of random variable $a$. We write $\mathcal{N}(\mu, \Sigma)$ as the Gaussian distribution with mean $\mu \in \mathbb{R}^d$ and covariance matrix $\Sigma \in \mathbb{R}^{d \times d}$. We let $\mathbb{S}^{d-1}$ be the unit sphere embedded in $\mathbb{R}^d$. We use $n, d$ to denote width and depth respectively, and write $G_n^{(d)}$ to stress $G$ has depth $d$ and width $n$, where $G$ can represent either a kernel or a neural network. We use the term *empirical NTK* to represent $\left\langle \frac{\partial f_n^{(d)}(\theta, x)}{\partial\theta}, \frac{\partial f_n^{(d)}(\theta, y)}{\partial\theta} \right\rangle$. We write $G^{(d)} = \lim_{n\to\infty} G_n^{(d)}$, $G_n = \lim_{d\to\infty} G_n^{(d)}$, and $G = \lim_{n,d\to\infty} G_n^{(d)}$ to denote limits are taken. All missing proofs can be found in the appendix.

## 3 NTK-OF-DEQ WITH FULLY-CONNECTED LAYERS

In this section, we show how to derive the NTK of the fully-connected DEQ.

Let $m$ be the input dimension, $x, y \in \mathbb{S}^{m-1}$ be a pair of inputs, $n$ be the width of the $h$-th layers where $h \in [d]$. Let $g^{(0)}(x) = \mathbf{0} \in \mathbb{R}^n$. Define the depth-$d$ approximation to a DEQ as the following:

$$f_n^{(h)}(x) = \sqrt{\frac{\sigma_W^2}{n}} W^{(h)} g^{(h-1)}(x) + \sqrt{\frac{\sigma_U^2}{n}} U^{(h)} x + \sqrt{\frac{\sigma_b^2}{n}} b^{(h)},$$
$$g_n^{(h)}(x) = \sigma(f^{(h)}(x)), \ f_n^{(d+1)}(x) = \sigma_v \cdot v^T g^{(d)}(x),$$

where $h \in [d]$, $W^{(h)} \in \mathbb{R}^{n \times n}$, $U^{(h)} \in \mathbb{R}^{n \times m}$, $v \in \mathbb{R}^n$ are the internal weights and $b^{(h)} \in \mathbb{R}^n$ are the bias terms.

The actual DEQ effectively outputs $f_n^{(\infty)} = \sigma_v \cdot v^T g_n^{(\infty)}(x) := \sigma_v \cdot v^T \left( \lim_{d\to\infty} g_n^{(d)}(x) \right)$. The forward pass is solved using root-finding or fixed point iteration, and the backward gradient is calculated using implicit function theorem instead of backpropogation.

One thing to note is that usually DEQs require tied-weights: $W^{(h)} = W$, $U^{(h)} = U$. and $b^{(h)} = b$ for all $h$. It turns out for the infinite width regime, DEQ with tied weights and DEQ without tied weights will induce the same NTK. We will discuss this point in more detail later.

Let $\Theta_n^{(d)}(x, y)$ be the empirical NTK of $f_n^{(d)}$. In Section 3.1, we will derive for an arbitrarily fixed $d$, the "finite depth iteration to DEQ-of-NTK" $\Theta^{(d)} = \lim_{n\to\infty} \Theta_n^{(d)}$. In Section 3.2, we show that $\Theta^{(d)}$ converges to a deterministic DEQ-of-NTK. Furthermore, we prove that $\lim_{d\to\infty} \lim_{n\to\infty} \Theta_n^{(d)} = \lim_{n\to\infty} \lim_{d\to\infty} \Theta_n^{(d)}$ with high probability, that is, the DEQ-of-NTK equals the NTK-of-DEQ.

## 3.1 FINITE DEPTH ITERATION TO DEQ-OF-NTK

Under the expressions in the beginning of Section 3, let us pick $\sigma_W, \sigma_U, \sigma_b \in \mathbb{R}$ arbitrarily in this section, and require the following NTK initialization.

**NTK initialization.** We randomly initialize every entry of every $W, U, b, v$ from $\mathcal{N}(0, 1)$.

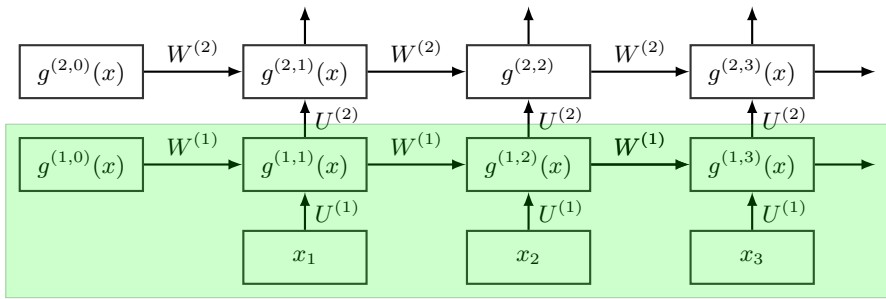

Figure 1: Visualization of a simple RNN from Alemohammad et al. (2020). The green area highlights a DEQ, if $x_1, x_2, \dots$ are all equal.

The finite depth iteration to the DEQ-of-NTK can be expressed as the following:

**Theorem 3.1.** *Recursively define the following quantities for $h \in [d]$:*

$$\Sigma^{(0)}(x,y) = x^\top y \tag{2}$$

$$\Lambda^{(h)}(x,y) = \begin{pmatrix} \Sigma^{(h-1)}(x,x) & \Sigma^{(h-1)}(x,y) \\ \Sigma^{(h-1)}(y,x) & \Sigma^{(h-1)}(y,y) \end{pmatrix} \tag{3}$$

$$\Sigma^{(h)}(x,y) = \sigma_W^2 \underset{\substack{(u,v)\sim \\ \mathcal{N}(0,\Lambda^{(h)})}}{\mathbb{E}} [\sigma(u)\sigma(v)] + \sigma_U^2 x^\top y + \sigma_b^2 \tag{4}$$

$$\dot{\Sigma}^{(h)}(x,y) = \sigma_W^2 \underset{\substack{(u,v)\sim \\ \mathcal{N}(0,\Lambda^{(h)})}}{\mathbb{E}} [\dot\sigma(u)\dot\sigma(v)] \tag{5}$$

$$\Sigma^{(d+1)}(x,y) = \sigma_v^2 \underset{\substack{(u,v)\sim \\ \mathcal{N}(0,\Lambda^{(h)})}}{\mathbb{E}} [\sigma(u)\sigma(v)] \tag{6}$$

$$\dot{\Sigma}^{(d+1)}(x,y) = \sigma_v^2 \underset{\substack{(u,v)\sim \\ \mathcal{N}(0,\Lambda^{(h)})}}{\mathbb{E}} [\dot\sigma(u)\dot\sigma(v)] \tag{7}$$

*Then the $d$-depth iteration to the DEQ-of-NTK can be expressed as:*

$$\Theta^{(d)}(x,y) = \sum_{h=1}^{d+2} \left( \left( \Sigma^{(h-1)}(x,y) \right) \cdot \prod_{h'=h}^{d+2} \dot{\Sigma}^{(h')}(x,y) \right), \tag{8}$$

*where by convention we set $\dot{\Sigma}^{(d+2)}(x,y) = 1$.*

One can realize that the derivation is done as if the weights in each layers are independently drawn from the previous layers, thus violating the formulation of DEQs. Nonetheless, it has been proven that under certain conditions, the tied-weight NN and untied-weight NN induce the same NTK, see Remark 3.2.

*Remark* 3.2. While our derivation is done on untied weights, the NTK of its weight-tying counterpart converges to the same point. This is formally done using the Nestor program introduced in Yang (2019; 2020). The neural architecture needs to satisfy a gradient independent assumption. One simple check is that the output layer weights are drawn from a zero-mean Gaussian independently from any other parameters and not used anywhere in the interior of the network. This is clearly satisfied in our setting. In fact, Alemohammad et al. (2020) has presented the recurrent NTK case with tied weights. Using their notation, by letting $g^{(1,0)}(\mathbf{x}) = \mathbf{0} \in \mathbb{R}^n$, $\mathbf{x}$ be $T$ copies of $x$, and $T = d$ represents the depth, we exactly recover the current (finite-depth) DEQ formulation. See Figure 1 for a visual explanation. Therefore, their conclusion directly applies to our setting. We should emphasize that our work is not a trivial extension to the recurrent NTK, because we mainly study the infinite-depth limit.

## 3.2 NTK-OF-DEQ EQUALS DEQ-OF-NTK

Based on Equation (8), we are now ready to show what the DEQ-of-NTK $\lim_{d\to\infty} \Theta^{(d)}$ is. Then we present the main takeaway of our paper: $\lim_{d\to\infty} \Theta^{(d)} = \lim_{n\to\infty} \lim_{d\to\infty} \Theta_n^{(d)}$. By convention, we assume the two samples $x, y \in \mathbb{S}^{d-1}$, and we require the parameters $\sigma_W^2, \sigma_U^2, \sigma_b^2$ obey the following DEQ-NTK initialization:

**DEQ-NTK initialization.** Let every entry of every $W, U, b, v$ follows the NTK initialization described in Section 3.1, as well as the additional requirement $\sigma_W^2 + \sigma_U^2 + \sigma_b^2 = 1$.

Let the nonlinear activation function $\sigma$ be the normalized ReLU: $\sigma(x) = \sqrt{2}\max(0, x)$ from now on.

Using normalized ReLU along with DEQ-NTK initialization, we can derive the main convergence theorem:

**Theorem 3.3.** *Use same notations and settings in Theorem 3.1, the DEQ-of-NTK is*

$$\Theta(x, y) \triangleq \lim_{d \to \infty} \Theta^{(d)}(x, y) = \frac{\sigma_v^2 \dot{\rho}^* \Sigma^*(x, y)}{1 - \dot{\Sigma}^*(x, y)} + \sigma_v^2 \rho^*, \tag{9}$$

*where $\Sigma^*(x, y) \triangleq \rho^*$ is the root of $R_\sigma(\rho) - \rho$,*

$$R_\sigma(\rho) \triangleq \sigma_W^2 \left( \frac{\sqrt{1 - \rho^2} + \left(\pi - \cos^{-1}\rho\right)\rho}{\pi} \right) + \sigma_U^2 x^\top y + \sigma_b^2, \tag{10}$$

*and*

$$\dot{\rho}^* \triangleq \left( \frac{\pi - \cos^{-1}(\rho^*)}{\pi} \right) \quad (11) \qquad \dot{\Sigma}^*(x, y) \triangleq \lim_{h \to \infty} \dot{\Sigma}^{(h)}(x, y) = \sigma_W^2 \dot{\rho}^*. \tag{12}$$

*Remark* 3.4. Note our $\Sigma^*(x, y)$ always depends on the inputs $x$ and $y$, so the information between two inputs is always preserved, even if the depth goes to infinity. On the contrary, as pointed out by Jacot et al. (2019), without input injection, $\Sigma^{(h)}(x, y)$ always converges to 1 as $h \to \infty$, even if $x \neq y$.

Theorem 3.3 provides us a way to direct calculate the DEQ-of-NTK by using root-finding algorithms. In practice, we can solve Equation (10) by using any optimization method. Then $\Sigma^*$ and $\Theta^*$ can be computed in constant time. Since each pair of input $(x, y)$ is independent of all the other pairs, we can easily parallelize this computation process. Our derivation can be extended to more complicated structures like DEQ with convolution layers, see appendix for more detail.

One caveat of Theorem 3.3 is the order of limits, notice that we first take the limit of the width, then the limit of the depth. Nonetheless, with sufficient conditions, one can indeed show that the limits can be exchanged, and the NTK-of-DEQ and the DEQ-of-NTK are equivalent.

**Theorem 3.5.** *Let $\sigma_W^2 \leq 1/8$, $\Theta_n^{(d)}(x, y) = \sum_{h=1}^{d+1} \left\langle \frac{\partial f(\theta, x)}{\partial \theta^{(h)}}, \frac{\partial f(\theta, y)}{\partial \theta^{(h)}} \right\rangle$ be the empirical NTK with depth $d$ and width $n$. Then $\lim_{n \to \infty} \lim_{d \to \infty} \Theta_n^{(d)} = \lim_{d \to \infty} \lim_{n \to \infty} \Theta_n^{(d)}$ ~~with high probability~~ in probability.*

*Proof sketch.* We first use a well-established random matrix result to conclude that $\sigma_W^2 < 1/8$ guarantees us that $\sigma \circ \sqrt{\sigma_W^2/n}W$ is a contraction with high probability. Using this contraction property, we can then show that the empirical NTK $\Theta_n^{(d)}$ converges. More importantly, it presents an "uniform convergence" property in $n$: a larger $d$ does not need a larger $n$ for the limit to converge. This is the crucial difference between this result and the results in untied-weight network. Intuitively, suppose contrarily our network has untied weights, to make our proof work we would need every layer's weight becomes a contraction. As $d$ increases, this clearly needs larger $n$ to use a union bound, which breaks if $d \to \infty$.

Finally, we prove a probabilistic version of Moore-Osgood theorem to conclude that our limit exchange result holds. □

*Remark* 3.6. In Theorem 3.5, for a fixed depth $d$, $\Theta^{(d)} := \lim_{n \to \infty} \Theta_n^{(d)}$ converges almost surely, hence we can view $\Theta := \lim_{d \to \infty} \Theta^{(d)}$ as a constant. On the other hand, for a fixed $n$, $\Theta_n := \lim_{d \to \infty} \Theta_n^{(d)}$ exists with probability at least $1 - e^{-c\epsilon^2 n}$ for some constant $c$, and $\epsilon \triangleq \frac{1 - 2\sqrt{2\sigma_W^2}}{\sqrt{2\sigma_W^2}}$. Formally, for any $\epsilon > 0$, we have

$$P\big(|\Theta_n - \Theta| > \epsilon\big) < o(n),$$

which converges in probability by definition.

*Remark* 3.7. We remark that Theorem 3.5 requires a more stringent $\sigma_W^2$ than Lemma B.1. This is indeed expected. For the actual DEQ to converge, one usually needs $I - W \succeq mI$ for some $m > 0$. It seems that $\sigma_W^2 \leq 1/2$ exactly reflects $I - W \succeq 0$, we leave this as an interesting future work. While Hanin & Nica (2019) also discussed about the relation between width and depth, and they concluded that the NTK may not even be deterministic if $d/n \gg 0$, our result does not contradict with theirs because their $n$ has to depend on $d$, but our proof decouples the dependency using uniform convergence thanks to weight-tying.

## 4 CASE STUDY: LINEAR DEQ

Theorem 3.5 shows a quite surprising result that we can safely exchange the limits, which is not at all straightforward to see. Consider the following linear DEQ case:

$$g_n^{(h)}(x) = \sqrt{\frac{\sigma_W^2}{n}} W g_n^{(h-1)}(x) + \sqrt{\frac{\sigma_U^2}{n}} U x, \ f_n^{(\infty)}(x) = v^T g^{(\infty)}(x). \tag{13}$$

Assuming the iteration converges (this can be guaranteed with high probability picking a suitable $\sigma_W$). Equivalently, we can also write this network as

$$f_n(x) = v^T \left( I - \sqrt{\frac{\sigma_W^2}{n}} W \right)^{-1} \sqrt{\frac{\sigma_U^2}{n}} U x. \tag{14}$$

Following the same derivation in Section 3, one can easily see that $\dot{\Sigma}^{(h)}(x, y) = \sigma_W^2$ for all $h$, and show that $\lim_{d\to\infty} \lim_{n\to\infty} \Theta_n^{(d)}(x, y) = \frac{\sigma_v^2 \sigma_U^2 x^T y}{(1-\sigma_W^2)^2} + \frac{\sigma_v^2 \sigma_U^2 x^T y}{1-\sigma_W^2}$. However, taking the infinite width limit of the network $f_n(x)$, it does not obey a Gaussian nature owing to the inverse of a shifted Gaussian matrix. It is not straightforward to see the limit exchange argument works. In this section, we aim to solve this linear DEQ case as a sanity check. In Section 5 we include numerical approximation that indicates the NTK-of DEQ-behaves as we expect.

**Theorem 4.1.** *Let $f_n(x)$ be defined as in Equation* (14) *and $\Theta_n^{(d)}$ be the empirical NTK associated with the finite depth approximation of $f_n$ in Equation* (13). *Let $\sigma_W^2 < 1/4$ and $\sigma_W^2 + \sigma_U^2 = 1$. We have*

$$\lim_{d\to\infty} \lim_{n\to\infty} \Theta_n^{(d)} = \lim_{n\to\infty} \lim_{d\to\infty} \Theta_n^{(d)} = \frac{\sigma_v^2 \sigma_U^2 x^T y}{(1 - \sigma_W^2)^2} + \frac{\sigma_v^2 \sigma_U^2 x^T y}{1 - \sigma_W^2}$$

*with high probability.*

*Proof sketch.* Let $H := \left( I - \sqrt{\frac{\sigma_W^2}{n}} W \right)^{-1}$. Such $H$ is well-defined with high probability if $\sigma_W^2 < 1/4$. A straightforward derivation gives:

$$\lim_{d\to\infty} \left\langle \frac{\partial f_n^{(d)}(x)}{\partial W}, \frac{\partial f_n^{(d)}(y)}{\partial W} \right\rangle = \frac{\sigma_U^2 \sigma_v^2}{n} \frac{\sigma_W^2}{n} \left\langle Hv(HUx)^T, Hv(HUx)^T \right\rangle$$

$$= \frac{\sigma_W^2 \sigma_U^2}{n} \langle HUx, HUx \rangle \frac{\sigma_v^2}{n} \langle Hv, Hv \rangle \xrightarrow{p} \sigma_U^2 \sigma_W^2 \sigma_v^2 x^T y \left( \frac{1}{n} \operatorname{tr}\left( H^T H \right) \right)^2 \tag{15}$$

$$\to \sigma_U^2 \sigma_W^2 \sigma_v^2 x^T y \left( \int \frac{1}{\lambda} d\mu(\lambda) \right)^2,$$

where the first convergence happens with high probability (Arora et al., 2019), and the second convergence holds for almost every realization of a sequence of $W$. This follows from the weak convergence of probability measure $\mu_n \xrightarrow{d} \mu$ a.s. and Portmanteau lemma, where $\mu_n$ is the empirical distribution of the eigenvalue of the matrix $\left( I - \sqrt{\frac{\sigma_W^2}{n}} W \right)^T \left( I - \sqrt{\frac{\sigma_W^2}{n}} W \right)$. More precisely, $\mu_n = \frac{1}{n} \sum_{i=1}^n \delta_{\lambda_i}$, $\delta_{\lambda_i}$ is the delta measure at the $i$th eigenvalue $\lambda_i$.

Next, we show that $\int \frac{1}{\lambda} d\mu(\lambda) = \frac{1}{1-\sigma_W^2}$. From Capitaine & Donati-Martin (2016), we learn that the Stieltjes transform $g$ of $\mu$ is a root to the following cubic equation:

$$\text{For } z \in \mathbb{C}^+ : g_\mu(z)^{-1} = \left(1 - \sigma_W^2 g_\mu(z)\right) z - \frac{1}{1 - \sigma_W^2 g_\mu(z)}.$$

We then apply the inverse formula of Stieltjes transformation to derive the density

$$d\mu(\lambda) = \frac{1}{\pi} \lim_{b \to 0^+} \text{Im } g_\mu(\lambda + ib). \tag{16}$$

This now involves a one-dimensional integration, which can be computed numerically and shown to be identical to the desired quantity. Similarly, we can compute that

$$\lim_{d \to \infty} \left\langle \frac{\partial f_n^{(d)}(x)}{\partial U}, \frac{\partial f_n^{(d)}(y)}{\partial U} \right\rangle \xrightarrow{p} \frac{\sigma_v^2 \sigma_U^2 x^T y}{1 - \sigma_W^2}, \quad \lim_{d \to \infty} \left\langle \frac{\partial f_n^{(d)}(x)}{\partial v}, \frac{\partial f_n^{(d)}(y)}{\partial v} \right\rangle \xrightarrow{p} \frac{\sigma_v^2 \sigma_U^2 x^T y}{1 - \sigma_W^2}.$$

Summing the three relevant terms and use the fact that $\sigma_U^2 + \sigma_W^2 = 1$, we get the claimed result. $\square$

## 5 SIMULATIONS

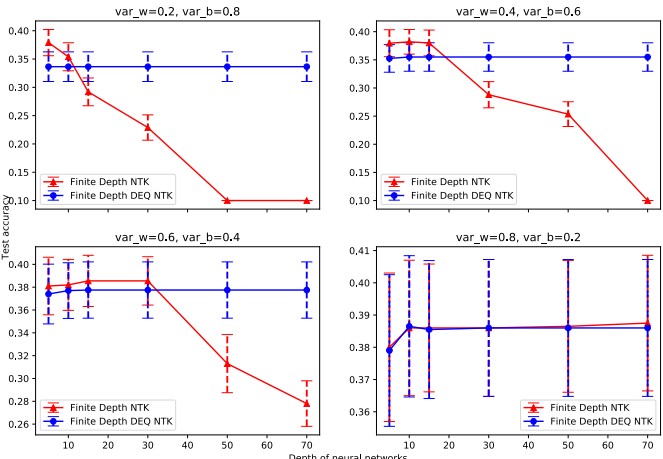

Figure 2: Finite depth NTK vs. finite depth iteration of NTK-of-DEQ. In all experiments, the NTK is initialized with $\sigma_W^2$ and $\sigma_b^2$ in the title. For NTK-of-DEQ we set $\sigma_U^2 = \sigma_b^2 - 0.1$ in the title, and $\sigma_b^2 = 0.1$. All models are trained on 1000 CIFAR-10 data and tested on 100 test data for 20 random draws. The error bar represents the $95\%$ confidence interval (CI). As expected, as the depth increases, the performance of NTKs drop, eventually their $95\%$ CI becomes a singleton, yet the performance of DEQs stabilize. Also note with larger $\sigma_W^2$, the freezing of NTK takes more depths to happen.

In this section, we perform numerical simulations on both synthetic data and real-world datasets including MNIST and CIFAR-10 to demonstrate our arguments. In particular, we show that (a) The NTK-of-DEQ and DEQ-of-NTK coincides, for both linear and non-linear cases, (b) A vanilla NTK of FCNN is degenerate while the NTK-of-DEQ escapes the freeze vs. chaos scheme, (c) The NTK-of-DEQ delivers reasonable performances on real-world datasets as a further evidence to its nondegeneracy.

### 5.1 NTK-OF-DEQ VS DEQ-OF-NTK

Recall in Section 4, the distribution $\mu$ in Equation (16) is that of the eigenvalues of $H^{-T} H^{-1} \triangleq (I - \sqrt{\sigma_W^2/n} W)^T (I - \sqrt{\sigma_W^2/n} W)$ as $n \to \infty$. The exact limiting eigenvalue distribution $\mu$ when

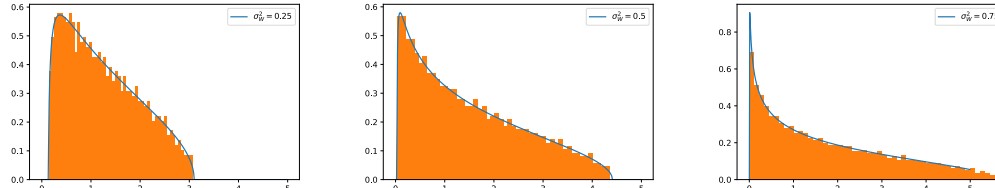

Figure 3: The empirical eigenvalue distribution of an instance of a $1000 \times 1000$ random matrix $(I - \sqrt{\sigma_W^2/n}W)^T(I - \sqrt{\sigma_W^2/n}W)$ with $\sigma_W^2 = 0.25, 0.5, 0.75$, respectively.

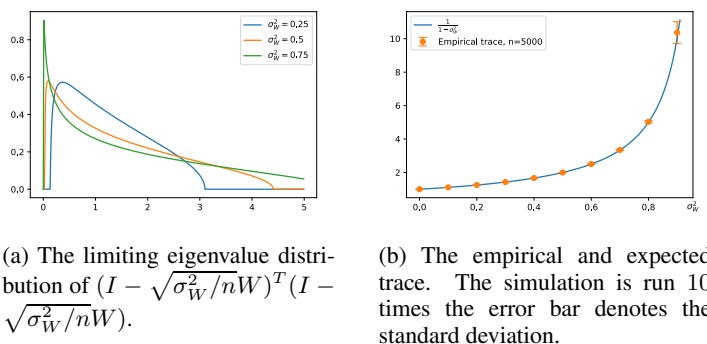

(a) The limiting eigenvalue distribution of $(I - \sqrt{\sigma_W^2/n}W)^T(I - \sqrt{\sigma_W^2/n}W)$.

(b) The empirical and expected trace. The simulation is run 10 times the error bar denotes the standard deviation.

Figure 4: Demonstrations of the limiting eigenvalue distribution of $H^{-T}H^{-1}$ and its approximation.

$\sigma_W^2 = 0.25, 0.5, 0.75$ is shown in Figure 4a. Keep in mind that $d\mu$ depicts the probability density of how large an eigenvalue of our random matrix can be.

For $\sigma_W^2 = 0.25, 0.5, 0, 75$ we include an empirical eigenvalue distribution of $H^{-T}H^{-1} \in \mathbb{R}^{n \times n}$ for $n = 1000$ in Figure 3. One can see that the empirical density is sufficiently close to the limiting distribution for large enough $n$, verifying the computation in Equation (16).

We calculated the empirical trace of $\frac{1}{n}\operatorname{tr} H^T H$ where $H$ is of size $5000 \times 5000$. This expression is the key element for Equation (15). The simulation samples $H$ i.i.d 10 times and the results are presented in Figure 4b. We can see that the variance of the estimator $1/(1 - \sigma_W^2)$ is negligible for small $\sigma_W^2$. Note that in the proof we require that $\left\| \sqrt{\sigma_W^2/n}W \right\| < 1$ with high probability, which holds when $\sigma_W^2 < 1/4$. However, empirically the convergence of empirical trace holds for much larger $\sigma_W^2$ as well.

We also test the difference between the empirical NTK-of-DEQ $\Theta_n$ and the DEQ-of-NTK $\Theta$ numerically, for both linear DEQ and nonlinear DEQ with normalized ReLU. We initialize both networks at variable width, with $\sigma_v^2 = 2$, $\sigma_W^2 = 1/8$, and $\sigma_U^2 = 7/8$. $\Theta_n$ is calculated by taking the inner product between the exact gradients[2] of a finite-width DEQ on two inputs, and $\Theta$ is computed using the DEQ-of-NTK formula in Theorem 3.3. A pair of input $(x, y)$ is randomly sampled and fixed throughout the simulation. For each width $n$, 10 trials are run, and we draw the mean of $\log \frac{|\Theta - \Theta_n|}{\Theta}$ in Figure 5. The convergence of the relative residue indicates that the NTK-of-DEQ and the DEQ-of-NTK coincide as proven.

## 5.2 SIMULATIONS ON CIFAR-10 AND MNIST

**Hyperparameter sensitivity.** We have three tunable parameters: $\sigma_W^2, \sigma_U^2, \sigma_b^2$. We try three random combinations listed in Table 3. As the results suggest, the performances of NTK-of-DEQ are insensitive to these parameters. This observation aligns with the description in Lee et al. (2020).

---

[2]The gradient is taken via implicit function theorem, see details in Bai et al. (2019).

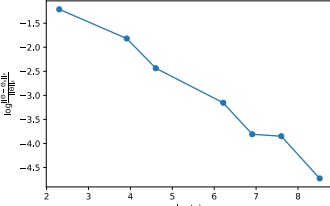 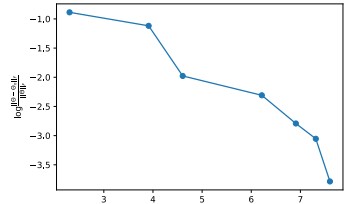

Figure 5: The deviation between the empirical NTK-of-DEQ and the exact DEQ-of-NTK on a log scale. The result of linear DEQ is on the left and the result of nonlinear DEQ is on the right. We randomly sample one pair of $(x, y)$ on the unit sphere, and for each width $n$, 10 trials are done with freshly sampled network weights, then we record the mean of relative residues in each setting. The convergence shows that NTK-of-DEQ and DEQ-of-NTK coincide.

Table 1: Performance of NTK-of-DEQ on MNIST and CIFAR-10 dataset.

| Parameters | Dataset | Acc. |
|---|---|---|
| $\sigma_W^2 = \sigma_U^2 = 0.25, \sigma_b^2 = 0.5$ | CIFAR-10 | 59.08% |
| $\sigma_W^2 = 0.6, \sigma_U^2 = 0.4, \sigma_b^2 = 0$ | CIFAR-10 | 59.77% |
| $\sigma_W^2 = 0.8, \sigma_U^2 = 0.2, \sigma_b^2 = 0$ | CIFAR-10 | 59.43% |
| $\sigma_W^2 = 0.6, \sigma_U^2 = 0.4$ | MNIST | 98.6% |

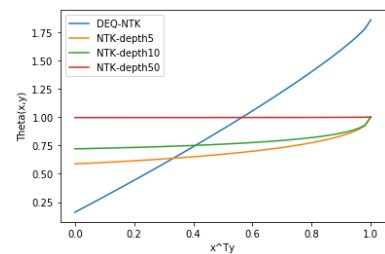

Figure 6: Relation between $\Theta(x, y)$ and $x^T y$.

**Training details and results.** For NTK-of-DEQ, following the theory, we normalize the dataset such that each data point has unit length. The fixed point $\Sigma^*(x, y)$ is solved by using the modified Powell hybrid method (Powell, 1970). Notice these root finding problems are one-dimensional, hence can be quickly solved.

After obtaining the NTK matrix, we apply kernel regressions (without regularization unless stated otherwise). For any label $y \in \{1, \ldots, n\}$, denote its one-hot encoding by $\mathbf{e}_y$. Let $\mathbf{1} \in \mathbb{R}^n$ be an all-1 vector, we train on the new encoding $-0.1 \cdot \mathbf{1} + \mathbf{e}_y$. That is, we change the "1" to 0.9, and the "0" to $-0.1$, as suggested by Novak et al. (2018). The results are listed in Table 3. These results prove that the NTK-of-DEQ is indeed non-degenerate.

On a smaller dataset with 1000 training data and 100 test data from CIFAR-10, we evaluate the performance of NTK and the finite depth iteration of NTK-of-DEQ, as depth increases. See Figure 2. When the depth increases, the performance of finite depth NTK gradually drops, eventually to 0.1 with 0 standard deviation. Also with larger $\sigma_W^2$, the degeneration of NTK occurs slower. This shows that large $\sigma_W^2$ preserves information from previous layers. Figure 6 also shows that the vanilla NTK becomes independent of the input inner product $x^T y$ as the depth increases. As proven in Jacot et al. (2019), the NTK will always "freeze" using the sets of parameters in Figure 2. In this scenario, the NTK Gram matrix becomes linearly independent as the depth increases, and its kernel regression does not have a unique solution. To circumvent this unsolvability, we add a regularization term $r \propto \frac{\epsilon \Theta(x,x)}{n}$, where $n$ is the size of the training data.

# 6 CONCLUSION

We derive NTKs for DEQ models, and show that they can be computed efficiently via root-finding based on a limit exchanging argument. This argument is proven theoretically for non-linear DEQs and an extra sanity check is done on linear DEQs, exploiting random matrix theory. Numerical simulations are performed to demonstrate that the limit exchanging phenomenon holds for both linear and non-linear NTK-of-DEQs. Our analysis also shows that one can avoid the freeze and chaos phenomenon in infinitely deep NTKs by using input injection. Additions experiments are

conducted to show that NTK-of-DEQs are non-degenerate on real-world datasets, while finite depth NTKs gradually degenerate as their depth increases.

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

## A  FORMAL DERIVATION OF WEIGHT-TIED NETWORK

In this section we formally derive the NTK of a DEQ (weight-tied) model, and show that they converge to the same limit as derived in Section 3. The argument is nearly identical to that of Alemohammad et al. (2020), which heavily depends on the NESTER⊤ program (Yang, 2020). We will first give a brief introduction, and then adapt to our setting.

**Definition A.1.** NESTER⊤ program is a program (as in type system) of which the variables take three-types: **A**-vars, **G**-vars, and **H**-vars. Any variables are generated by one of the rules in `MatMul` (matrix multiplication), `NonLin` (nonlinearity), `LinComb` (linear combination), or `Trsp` (matrix transpose). We also sometimes explicitly express the dimensionality of a variable in the following way:

- If $x \in \mathbb{R}^n$, and is of type **G**, **H**, we write $x : \mathbf{G}(n)$ or $x : \mathbf{H}(n)$.

- If $A \in \mathbb{R}^{n \times m}$, we write $A : \mathbf{A}(n, m)$.

The program goes as following:

**Input**   A set of **G**-vars and **A**-vars.

**Body**   Any variable is introduced by the following rules:

- `Trsp`. If $A : \mathbf{A}(n, m)$, then $A^\top : \mathbf{A}(m, n)$.

- `MatMul`. If $A : \mathbf{A}(n, m)$ and $x : \mathbf{H}(m)$, then $Ax : \mathbf{G}(n)$.

- `LinComb`. If $g^1, \ldots, g^k : \mathbf{G}(n)$ and $a^1, \ldots, a^k \in \mathbb{R}$, then $\sum_{i=1}^k a^i g^i : \mathbf{G}(n)$.

- `NonLin`. If $x^1, \ldots, x^k : \mathbf{G}(n)$, and $\phi : \mathbb{R}^k \to \mathbb{R}$ is a coordinate-wise nonlinear function, then $\phi(x^1, \ldots, x^k) : \mathbf{H}(n)$.

**Output**   The program outputs a scalar of the form

$$\frac{1}{n} \sum_{\alpha=1}^n \psi \left( h_\alpha^1, \ldots, h_\alpha^k \right)$$

for $h^1 \ldots h^k : \mathbf{H}(n)$.

For example, a depth-$d$ approximation to a DEQ model is provided in Algorithm 1. For simplicity, we left out the scaling $\sigma_W^2 / \sqrt{n}$ (as was done in Yang (2020)).

---

**Algorithm 1 NESTER⊤ program** Depth-$d$ approximation to a DEQ model

---

**Require:** $Ux, Uy : \mathbf{G}(n), W : \mathbf{A}(n, n), b : \mathbf{G}(n), v : \mathbf{G}(n)$. Polynomially-bounded coordinate-wise nonlinear function $\phi$.

  **for** $h = 1, \ldots, d$ **do**
    **for** $z \in \{x, y\}$ **do**
      $f^{(h)}(z) = W g^{(h-1)}(z) + Uz + b : \mathbf{G}(n)$.
      $g^{(h)}(z) = \phi(f^{(h)}(z)) : \mathbf{H}(n)$.
      // The network outputs $f^{(d+1)}(z) := \frac{v^\top g^{(d)}(z)}{n}$, but we don't express this in the program.
      // Backprop, for varible $u$, let $du := \sqrt{n} \nabla_u f^{(d+1)}(z)$.
      $dg^{(d)}(z) = v : \mathbf{G}(n)$.
      $df^{(d)}(z) = \phi'(f^{(d)}(z)) \odot dg^{(d)}(z) : \mathbf{H}(n)$.         ▷ We use $\odot$ for Hadamard product.
      $dg^{(h)}(z) = W^\top df^{(h+1)}(z) : \mathbf{G}(n)$.
      $df^{(h)}(z) = \phi'(f^{(h)}(z)) \odot dg^{(h)}(z) : \mathbf{H}(n)$.
    **end for**
  **end for**

---

One can express many neural network architectures into a NESTER$\top$ program, but not all. The required regularity condition is the so-called *BP-like*:

**Definition A.2** (BP-like). A NESTER$\top$ program is *BP-like* if there exists a non-empty set of input $\mathbf{G}(n)$-vars $v^1, \ldots, v^k$ s.t:

1. If $W^\top z$ is used in the program for some $z : \mathbf{H}(n)$, and $W : \mathbf{A}(n, m)$ is an input $\mathbf{A}$-var, then $z$ must be an odd function of $v^1, \ldots, v^k$. That is,

$$z\left(-v^1, \ldots, -v^k, \text{all other } \mathbf{G}\text{-vars}\right) = -z\left(v^1, \ldots, v^k, \text{all other } \mathbf{G}\text{-vars}\right).$$

2. If $Wz$ is used in the program for some $z : \mathbf{H}(m)$, and $W : \mathbf{A}(n, m)$ is an input $\mathbf{A}$-var, then $z$ cannot depend on any of $v^1, \ldots v^k$.

3. $v^1, \ldots, v^k$ are sampled with zero mean and independently from all other $\mathbf{G}$-vars.

**Definition A.3** (Polynomially-bounded). We say a function $f : \mathbb{R}^k \to \mathbb{R}$ is polynomially-bounded if $|\phi(x)| \leq C\|x\|^p + c$ for some $c, C, p > 0$, for all $x \in \mathbb{R}^k$. Note that ReLU and inner product are polynoimially-bounded.

Recall that the simple gradient independence assumption (GIA) check we give in Section 3:

**Condition A.4** (Simple GIA check). *Gradient independence assumption is a heuristic that for any matrix $W$, we assume $W^\top$ used in backprop is independet from $W$ used in the forward pass. We can regard this assumption holds in the NTK computation if the following simple check holds: the output layer is sampled independently with zero mean from all other parameters and it not used anywhere else in the interior of the network, that is, if the output of the network is $v^\top x$, then $v$ is independent of $x$.*

Apparently our DEQ formulation satisfy the simple GIA check, notice that by formulation, the second and third condition in Definition A.2 are trivially satisfied. Also since $v$ is the last layer weight, any $\mathbf{G}$-var of the form $W^\top z$ only shows up in the backpropagation, and is linear (thus odd) in $v$ as well. Hence the first condition is also satisified. So any network structure that satisfies the simple GIA check is automatically BP-like.

**Setup A.5.** *For NESTER$\top$ program, we assume that each entry in $W : \mathbf{A}(n, m)$ is sampled from $\mathcal{N}(0, \sigma_W^2/n)$, and any input $\mathbf{G}$-vars $x \sim \mathcal{N}(\mu^{in}, \Sigma^{in})$. We remark that this does not contradict with the parameterization that we mentioned in the main text where the entries of input $\mathbf{A}$-vars $W, U$ are standard Gaussians. One just needs to properly scale their variables.*

**Theorem A.6** (BP-like NESTER$\top$ program Master theorem). *Fix any BP-like NESTER$\top$ program that satisfies Setup A.5, and all its nonlinearities are polynomially-bounded. If $g^1, \ldots, g^M$ are all $\mathbf{G}$-vars in the program, then for any polynomially-bounded $\psi : \mathbb{R}^M \to \mathbb{R}$, as $n \to \infty$, we have*

$$\frac{1}{n} \sum_{\alpha=1}^n \psi\left(g_\alpha^1, \ldots, g_\alpha^M\right) \xrightarrow{a.s.} \mathop{\mathbb{E}}_{Z \sim \mathcal{N}(\mu, \Sigma)} \psi(Z) = \mathop{\mathbb{E}}_{Z \sim \mathcal{N}(\mu, \Sigma)} \psi\left(Z^{g^1}, \ldots, Z^{g^M}\right),$$

*where $Z = \{Z^{g^1}, \ldots, Z^{g^M}\} \in \mathbb{R}^M$, $\mu = \{\mu(g^i)\}_{i \in [M]} \in \mathbb{R}^M$, $\Sigma = \{\Sigma(g^i, g^j)\}_{i,j=1}^M \in \mathbb{R}^{M \times M}$ are given by*

$$\mu(g) = \begin{cases} \mu^{in}(g) & \text{if } g \text{ is input,} \\ \sum_{i=1}^k a^i \mu(g^i) & \text{if } g = \sum_{i=1}^k a^i g^i \\ 0 & \text{otherwise} \end{cases}$$

$$\Sigma(g, \bar{g}) = \begin{cases} \Sigma^{in}(g, g') & \text{if } g, g' \text{ are inputs} \\ \sum_{i=1}^k a^i \Sigma(g^i, \bar{g}) & \text{if } g = \sum_{i=1}^k a^i g^i \\ \sum_{i=1}^k a^i \Sigma(g, \bar{g}^i) & \text{if } \bar{g} = \sum_{i=1}^k a^i \bar{g}^i \\ \sigma_W^2 \mathbb{E}_Z \phi(Z) \bar{\phi}(Z) & \text{if } g = Wh, \bar{g} = W\bar{h}, \\ 0 & \text{otherwise.} \end{cases} \tag{17}$$

We are now equipped to derive the NTK of a depth-$d$ approximation to a DEQ. Particularly, we have

$$\nabla_W f^{(d+1)}(x) = \frac{\sigma_W}{n} \sum_{h=1}^d df^{(h)} g^{(h-1)}(x)^\top,$$

hence

$$\left\langle \nabla_W f^{(d+1)}(x), \nabla_W f^{(d+1)}(y) \right\rangle = \sigma_W^2 \sum_{l,h=1}^{d} \frac{df^{(h)}(x)^\top df^{(l)}(y)}{n} \frac{g^{(h-1)}(x)^\top g^{(l-1)}(y)}{n}.$$

From this point, we need to calculate

$$\mathbb{E}_W \left[ df^{(h)}(x)^\top df^{(l)}(y) \right] \text{ and } \mathbb{E}_W \left[ g^{(h-1)}(x)^\top g^{(l-1)}(y) \right].$$

In the end, applying the Master theorem with $\psi(x,y) = x^\top y$ on $\frac{df^{(h)}{}^\top df^{(l)}}{n}$ and $\frac{g^{(h-1)}(x)^\top g^{(l-1)}(y)}{n}$ shows that these empirical averages converge to the expectations.

*Remark* A.7. Notice that the Master theorem talks about **G**-vars, while $df^{(h)}$ and $g^{(h)}$ are **H**-vars. We can always compose $\psi' = \psi \circ \phi$, where $\psi$ is the inner product and $\phi$ is coordinate-wise nonlinearity (such as ReLU), and apply the Master theorem on $\psi'$, as long as it is still polynomially-bounded.

$$\mathbb{E}_W \left[ df^{(h)}(x)^\top df^{(l)}(y) \right] = \mathbb{E} \left[ \left( \phi'(f^{(h)}(x)) \odot dg^{(h)}(x) \right)^\top \left( \phi'(f^{(l)}(y)) \odot dg^{(l)}(y) \right) \right]$$

$$= \mathbb{E} \left[ \phi'(f^{(h)}(x))^\top \phi'(f^{(l)}(y)) \cdot (dg^{(h)}(x)^\top dg^{(l)}(y)) \right]$$

$$= \underbrace{\mathbb{E} \left[ \phi'(f^{(h)}(x))^\top \phi'(f^{(l)}(y)) \right]}_{A} \cdot \underbrace{\mathbb{E} \left[ (dg^{(h)}(x)^\top dg^{(l)}(y)) \right]}_{B}.$$

By the Master theorem and GIA, $\phi'(f^{(h)})$ and $dg^{(h)}$ are introduced by different **A**-vars ($W$ and $W^\top$), hence their coviance is 0. This justifies the last step above.

When $h, l < d$, by the Master theorem we have

$$B = \sigma_W^2 \mathbb{E}[df^{(h+1)}(x)^\top df^{(l+1)}(y)].$$

Notice that this gives a recursive expression, WLOG we assume that $h < l$, this induction will lead to

$$\mathbb{E}[df^{(h+t)}(x)^\top df^{(d)}(y)] = \mathbb{E} \left[ \left( \phi'(f^{(h+t)}(x)) \odot dg^{(h+t)}(x) \right)^\top \left( \phi'(f^{(d)}(y)) \odot v \right) \right] = 0,$$

for some $t > 0$. The reason why this is zero is still due to the Master theorem, as $df^{(h+t)}(x)$ and $df^{(d)}(y)$ are **G**-vars involved with different **A**-vars $W$ and $v$.

This shows that when $h \neq l$, $\mathbb{E}_W \left[ df^{(h)}(x)^\top df^{(l)}(y) \right] = 0$. Hence we only have to consider the case $h = l$. By the Master theorem we have

$$A = \mathbb{E}_{u,v} \left[ \phi'(u)\phi'(v) \right], \mathbb{E}_W \left[ g^{(h)}(x)^\top g^{(h)}(y) \right] = \mathbb{E}_{u,v} \left[ \phi(u)\phi(v) \right],$$

where

$$(u,v) \sim \mathcal{N} \left( 0, \begin{pmatrix} \Sigma^{(h-1)}(x,x) & \Sigma^{(h-1)}(x,y) \\ \Sigma^{(h-1)}(y,x) & \Sigma^{(h-1)}(y,y) \end{pmatrix} \right).$$

Notice this exactly recovers the calculation of NTK when the weights are un-tied. The exact same argument can be applied to $\nabla_U f$ and $\nabla_b f$. Since such equivalence holds for all depth $d$, it also holds in the limit of $d \to \infty$.

**Key takeaway** The NESTER$\top$ program allows us to calculate the NTK of a weight-tied network in exactly the same way as the weight-untied network.

# B    DETAILS OF SECTION 3

In this section, we give the detailed derivation of DEQ-of-NTK. There are two terms that are different from NTK: $\Sigma^{(h)}(x,y)$ and the extra $\mathbb{E}_\theta \left[ \left\langle \frac{\partial f(\theta,x)}{\partial U}, \frac{\partial f(\theta,y)}{\partial U} \right\rangle \right]$ in the kernel.

Let us restate the depth-$d$ approximation to DEQs here:

Let $m$ be the input dimension, $x, y \in \mathbb{R}^m$ be a pair of inputs, $n$ be the width of the $h^{th}$ hidden layers. Define the depth-$d$ approximation to DEQ as follows:

$$f_\theta^{(h)}(x) = \sqrt{\frac{\sigma_W^2}{n}} W^{(h)} g^{(h-1)}(x) + \sqrt{\frac{\sigma_U^2}{n}} U^{(h)} x + \sqrt{\frac{\sigma_b^2}{n}} b^{(h)}, \; h \in [L]$$

$$g^{(d)}(x) = \sigma(f_\theta^{(L)}(x))$$

$$f^{(d+1)}(x) = \sigma_v^2 \cdot v^T g_\theta^{(d+1)}(x)$$

where $W^{(h)} \in \mathbb{R}^{n \times n}$, $U^{(h)} \in \mathbb{R}^{n \times m}$, and $v \in \mathbb{R}^n$ are the internal weights, and $b^{(h)} \in \mathbb{R}^n$ are the bias terms. These parameters are chosen using the NTK initialization. Let us pick $\sigma_W, \sigma_U, \sigma_b \in \mathbb{R}$ arbitrarily in this section.

**Theorem 3.1.** *Recursively define the following quantities for $h \in [d]$:*

$$\Sigma^{(0)}(x,y) = x^\top y \qquad (2)$$

$$\Lambda^{(h)}(x,y) = \begin{pmatrix} \Sigma^{(h-1)}(x,x) & \Sigma^{(h-1)}(x,y) \\ \Sigma^{(h-1)}(y,x) & \Sigma^{(h-1)}(y,y) \end{pmatrix} \qquad (3)$$

$$\Sigma^{(h)}(x,y) = \sigma_W^2 \mathop{\mathbb{E}}_{\substack{(u,v)\sim \\ \mathcal{N}(0,\Lambda^{(h)})}} [\sigma(u)\sigma(v)]$$
$$+ \sigma_U^2 x^\top y + \sigma_b^2 \qquad (4)$$

$$\dot\Sigma^{(h)}(x,y) = \sigma_W^2 \mathop{\mathbb{E}}_{\substack{(u,v)\sim \\ \mathcal{N}(0,\Lambda^{(h)})}} [\dot\sigma(u)\dot\sigma(v)] \qquad (5)$$

$$\Sigma^{(d+1)}(x,y) = \sigma_v^2 \mathop{\mathbb{E}}_{\substack{(u,v)\sim \\ \mathcal{N}(0,\Lambda^{(h)})}} [\sigma(u)\sigma(v)] \qquad (6)$$

$$\dot\Sigma^{(d+1)}(x,y) = \sigma_v^2 \mathop{\mathbb{E}}_{\substack{(u,v)\sim \\ \mathcal{N}(0,\Lambda^{(h)})}} [\dot\sigma(u)\dot\sigma(v)] \qquad (7)$$

*Then the $d$-depth iteration to the DEQ-of-NTK can be expressed as:*

$$\Theta^{(d)}(x,y) = \sum_{h=1}^{d+2} \left( \left( \Sigma^{(h-1)}(x,y) \right) \cdot \prod_{h'=h}^{d+2} \dot\Sigma^{(h')}(x,y) \right), \qquad (8)$$

*where by convention we set $\dot\Sigma^{(d+2)}(x,y) = 1$.*

*Proof of Theorem 3.1.* First we note that

$$\mathbb{E}\left[ \left[ f^{(h+1)}(x) \right]_i \cdot \left[ f^{(h+1)}(y) \right]_i \mid f^{(h)} \right]$$

$$= \frac{\sigma_W^2}{n} \sum_{j=1}^n \sigma\left( \left[ f^{(h)}(x) \right]_j \right) \sigma\left( \left[ f^{(h)}(y) \right]_j \right) + \frac{\sigma_U^2}{n} \sum_{j=1}^n x^\top y + \sigma_b^2$$

$$\to \Sigma^{(h+1)}(x,y) \; a.s$$

where the first line is by expansion the original expression and using the fact that $W, U, b$ are all independent. The last line is from the strong law of large numbers. This shows how the covariance changes as depth increases with input injection.

Recall the splitting:

$$\Theta^{(L)}(x,y) = \mathbb{E}_\theta \left[ \left\langle \frac{\partial f(\theta,x)}{\partial \theta}, \frac{\partial f(\theta,y)}{\partial \theta} \right\rangle \right]$$

$$= \underbrace{\mathbb{E}_\theta \left[ \left\langle \frac{\partial f(\theta,x)}{\partial W}, \frac{\partial f(\theta,y)}{\partial W} \right\rangle \right]}_{\text{①}} + \underbrace{\mathbb{E}_\theta \left[ \left\langle \frac{\partial f(\theta,x)}{\partial U}, \frac{\partial f(\theta,y)}{\partial U} \right\rangle \right]}_{\text{②}}$$

$$+ \underbrace{\mathbb{E}_\theta \left[ \left\langle \frac{\partial f(\theta,x)}{\partial b}, \frac{\partial f(\theta,y)}{\partial b} \right\rangle \right]}_{\text{③}} + \underbrace{\mathbb{E}_\theta \left[ \left\langle \frac{\partial f(\theta,x)}{\partial v}, \frac{\partial f(\theta,y)}{\partial v} \right\rangle \right]}_{\text{④}}$$

The following equation has been proven in many places:

$$\text{①} = \sum_{h=1}^{d+1} \left( \sigma_W^2 \underset{(u,v)\sim\mathcal{N}(0,\Lambda^{(h)})}{\mathbb{E}} [\sigma(u)\sigma(v)] \cdot \prod_{h'=h}^{d+1} \dot{\Sigma}^{(h')}(x,y) \right), \quad \text{③} = \sum_{h=1}^{d+1} \left( \sigma_b^2 \cdot \prod_{h'=h}^{d+1} \dot{\Sigma}^{(h')}(x,y) \right),$$

and $\text{④} = \sigma_v^2 \, \mathbb{E}_{(u,v)\sim\mathcal{N}(0,\Lambda^{(h)})}[\sigma(u)\sigma(v)]$. For instance, see Arora et al. (2019). So we only need to deal with the second term $\mathbb{E}_\theta \left[ \left\langle \frac{\partial f(\theta,x)}{\partial U}, \frac{\partial f(\theta,y)}{\partial U} \right\rangle \right]$. Write $f = f_\theta(x)$ and $\tilde{f} = f_\theta(y)$, by chain rule, we have

$$\left\langle \frac{\partial f}{\partial U^{(h)}}, \frac{\partial \tilde{f}}{\partial U^{(h)}} \right\rangle$$

$$= \left\langle \frac{\partial f}{\partial f^{(h)}} \frac{\partial f^{(h)})}{\partial U^{(h)}}, \frac{\partial \tilde{f}}{\partial \tilde{f}^{(h)}} \frac{\partial \tilde{f}^{(h)})}{\partial U^{(h)}} \right\rangle$$

$$= \left\langle \frac{\partial f^{(h)}}{\partial U^{(h)}}, \frac{\partial \tilde{f}^{(h)}}{\partial U^{(h)}} \right\rangle \cdot \left\langle \frac{\partial f}{\partial f^{(h)}}, \frac{\partial \tilde{f}}{\partial \tilde{f}^{(h)}} \right\rangle$$

$$\to \sigma_U^2 x^\top y \cdot \prod_{h'=h}^{d+1} \dot{\Sigma}^{(h')}(x,y)$$

where the last line uses the existing conclusion that $\left\langle \frac{\partial f}{\partial f^{(h)}}, \frac{\partial \tilde{f}}{\partial \tilde{f}^{(h)}} \right\rangle \to \prod_{h'=h}^{d+1} \dot{\Sigma}^{(h')}(x,y)$, this convergence almost surely holds when $N \to \infty$ by law of large numbers.

Finally, summing $\left\langle \frac{\partial f}{\partial U^{(h)}}, \frac{\partial \tilde{f}}{\partial U^{(h)}} \right\rangle$ over $h \in [d]$ we conclude the assertion. $\square$

**Lemma B.1.** *Use the same notations and settings in Theorem 3.1. With input data $x,y \in \mathbb{S}^{d-1}$, parameters $\sigma_W^2, \sigma_U^2, \sigma_b^2$ following the DEQ-NTK initialization, $\Theta^{(d)}(x,y)$ in Equation (8) converges absolutely if $\sigma_W^2 < 1$.*

*Proof.* Since we pick $x,y \in \mathbb{S}^{d-1}$, and by DEQ-NTK initialization, we always have $\Sigma^{(h)}(x,y) < 1$ for $x \neq y$. Let $\rho = \Sigma^{(h)}(x,y)$, by Equation (5) and Equation (19), if $\sigma_W^2 < 1$, then there exists $c$ such that $\dot{\Sigma}^{(h)}(x,y) < c < 1$ for finite number of pairs $x \neq y$ on $\mathbb{S}^{d-1}$, and large enough $h$. This is because $\lim_{h\to\infty} \dot{\Sigma}^{(h)}(x,y) = \dot{\Sigma}^*(x,y) < \dot{\Sigma}^*(x,x) < 1$.

Use comparison test,

$$\lim_{L\to\infty} \sum_{h=1}^{L+1} \left| \left( \Sigma^{(h-1)}(x,y) \right) \cdot \prod_{h'=h}^{L+1} \dot{\Sigma}^{(h')}(x,y) \right| < 1 + \lim_{L\to\infty} \sum_{h=1}^{L+1} c^{L+1-h}.$$

Since $c < 1$, the geometric sum converges absolutely, hence $\Theta^{(d)}(x,y)$ converges absolutely if $\sigma_W^2 < 1$, and the limit exists. $\square$

**Theorem 3.3.** *Use same notations and settings in Theorem 3.1, the DEQ-of-NTK is*

$$\Theta(x,y) \triangleq \lim_{d\to\infty} \Theta^{(d)}(x,y) = \frac{\sigma_v^2 \dot{\rho}^* \Sigma^*(x,y)}{1 - \dot{\Sigma}^*(x,y)} + \sigma_v^2 \rho^*, \tag{9}$$

*where $\Sigma^*(x,y) \triangleq \rho^*$ is the root of $R_\sigma(\rho) - \rho$,*

$$R_\sigma(\rho) \triangleq \sigma_W^2 \left( \frac{\sqrt{1-\rho^2} + \left(\pi - \cos^{-1}\rho\right)\rho}{\pi} \right) + \sigma_U^2 x^\top y + \sigma_b^2, \tag{10}$$

*and*

$$\dot{\rho}^* \triangleq \left( \frac{\pi - \cos^{-1}(\rho^*)}{\pi} \right) \tag{11} \qquad \dot{\Sigma}^*(x,y) \triangleq \lim_{h\to\infty} \dot{\Sigma}^{(h)}(x,y) = \sigma_W^2 \dot{\rho}^*. \tag{12}$$

*Proof of Theorem 3.3.* Due to the fact that $x \in \mathbb{S}^{d-1}$, $\sigma$ being normalized, and DEQ-NTK initialization, one can easily calculate by induction that for all $h \in [L]$: $\Sigma^{(h)}(x,x) = \sigma_W^2 \mathbb{E}_{u\sim\mathcal{N}(0,1)}[\sigma(u)^2] + \sigma_V^2 x^\top x + \sigma_b^2 = 1$ This indicates that in Equation (3), the covariance matrix has a special structure $\Lambda^{(h)}(x,y) = \begin{pmatrix} 1 & \rho \\ \rho & 1 \end{pmatrix}$, where $\rho = \Sigma^{(h-1)}(x,y)$ depends on $h, x, y$. For simplicity we omit the $h, x, y$ in $\Lambda^{(h)}(x,y)$. As shown in Daniely et al. (2016):

$$\mathbb{E}_{(u,v)\sim\mathcal{N}(0,\Lambda)} [\sigma(u)\sigma(v)] = \frac{\sqrt{1-\rho^2} + \left(\pi - \cos^{-1}(\rho)\right)\rho}{\pi} \tag{18}$$

$$\mathbb{E}_{(u,v)\sim\mathcal{N}(0,\Lambda)} [\dot{\sigma}(u)\dot{\sigma}(v)] = \frac{\pi - \cos^{-1}(\rho)}{\pi} \tag{19}$$

Adding input injection and bias, we derive Equation (10) from Equation (18), and similarly, Equation (12) from Equation (19). Notice that iterating Equations (2) to (4) to solve for $\Sigma^{(h)}(x,y)$ is equivalent to iterating $(R_\sigma \circ \cdots \circ R_\sigma)(\rho)$ with initial input $\rho = x^\top y$. Take the derivative

$$\left| \frac{dR_\sigma(\rho)}{d\rho} \right| = \left| \sigma_W^2 \left( 1 - \frac{\cos^{-1}(\rho)}{\pi} \right) \right| < 1, \text{ if } \sigma_W^2 < 1 \text{ and } -1 \le \rho < 1.$$

For $x \ne y$ we have $-1 \le \rho < c < 1$ for some $c$ (this is because we only have finite number of inputs $x, y$) and by DEQ-NTK initialization we have $\sigma_W^2 < 1$, so the above inequality hold. Hence $R_\sigma(\rho)$ is a contraction on $[0, c]$, and we conclude that the fixed point $\rho^*$ is attractive.

By Lemma B.1, if $\sigma_W^2 < 1$, then the limit of Equation (8) exists, so we can rewrite the summation form in Equation (8) in a recursive form:

$$\Theta^{(0)}(x,y) = \Sigma^{(0)}(x,y),$$
$$\Theta^{(d+1)}(x,y) = \dot{\Sigma}^{(d+1)}(x,y) \cdot \Theta^{(d)}(x,y) + \Sigma^{(d+1)}(x,y).$$

Directly solve the fixed point iteration for the internal representation:

$$\begin{aligned} &\lim_{d\to\infty} \Theta^{(d+1)}(x,y) \\ &= \lim_{d\to\infty} \left( \dot{\Sigma}^{(d+1)}(x,y) \cdot \Theta^{(d)}(x,y) + \Sigma^{(d+1)}(x,y) \right) \\ \Longrightarrow &\lim_{L\to\infty} \Theta^{(d+1)}(x,y) \\ &= \dot{\Sigma}^*(x,y) \cdot \lim_{d\to\infty} \Theta^{(d)}(x,y) + \Sigma^*(x,y) \\ \Longrightarrow &\lim_{d\to\infty} \Theta^{(d)}(x,y) \\ &= \dot{\Sigma}^*(x,y) \cdot \lim_{d\to\infty} \Theta^{(d)}(x,y) + \Sigma^*(x,y). \end{aligned} \tag{20}$$

Solving for $\lim_{d\to\infty} \Theta^{(d)}(x,y)$ we get $\Theta^*(x,y) = \frac{\Sigma^*(x,y)}{1-\dot\Sigma^*(x,y)}$. Finally, we process the classification layer and get $\Theta = \dot\Sigma \cdot \Theta^* + \Sigma$, where $\dot\Sigma = \sigma_v^2 \dot\rho^*$ and $\Sigma = \sigma_v^2 \rho^*$. This concludes the proof

$\square$

## B.1 DEQ-OF-NTK VS. NTK-OF-DEQ

In this section we discuss Theorem 3.5 in detail. Recall that the NTK is the kernel matrix formed by an infinitely-wide network. To be more precisely, if the network has depth $d$, then

$$\Theta^{(d)}(x,y) = \mathbb{E}_\theta \left[ \left\langle \frac{\partial f(\theta,x)}{\partial \theta}, \frac{\partial f(\theta,y)}{\partial \theta} \right\rangle \right].$$

It is straightforward to define its width-$n$ approximation:

$$\Theta_n^{(d)} = \sum_{h=1}^{d} \left\langle \frac{\partial f(\theta,x)}{\partial \theta^{(h)}}, \frac{\partial f(\theta,y)}{\partial \theta^{(h)}} \right\rangle,$$

where $\theta^{(h)}$ is the parameter of the $h$th layer with width $n$. The name of $\lim_{d\to\infty} \lim_{n\to\infty} \Theta_n^{(d)}$ being the DEQ of NTK is intuitive: because we first [3] bring width to infinity, that is, the NTK is first derived. Then we talk about the NTK's infinite-depth limit. This is in distinction to our desired quantity, $\lim_{n\to\infty} \lim_{d\to\infty} \Theta_n^{(d)}$, which is the NTK of DEQ naturally. In this section we show they are indeed equivalent under certain conditions.

First we introduce some notations. Consider a finite depth iteration of a NTK with depth $d+1$, and for simplicity let the bias term $b^{(h)} = 0$ for all $h \in [d+1]$. A straightforward calculation show that

$$\text{For } h \in [L+1]: \frac{df(\theta,x)}{dW^{(h)}} = p^{(h)}(x)\Big(g^{(h-1)}(x)\Big)^\top$$

$$\frac{df(\theta,x)}{dU^{(h)}} = p^{(h)}(x) \cdot x^\top$$

$$\text{where } p^{(h)}(x) = \begin{cases} 1 \in \mathbb{R}^n, & h = d+1 \\ \sqrt{\frac{\sigma_W^2}{N_h}} \operatorname{diag}\Big(\dot\sigma\Big(f^{(h)}(x)\Big)\Big)\Big(W^{(h+1)}\Big)^\top p^{(h+1)}(x) & h \le d \end{cases}$$

Here $\operatorname{diag}\Big(\dot\sigma\Big(f^{(h)}(x)\Big)\Big) \in \mathbb{R}^{N_h \times N_h}$. Let $N_h = n$ for all $h$, and $W^{(h+1)} := v$. Notice that

$$\operatorname{diag}\Big(\dot\sigma\Big(f^{(h)}(x)\Big)\Big)\Big(W^{(h+1)}\Big)^\top p^{(h+1)}(x) = \dot\sigma\Big(f^{(h)}(x)\Big) \odot \Big(\Big(W^{(h+1)}\Big)^\top p^{(h+1)}(x)\Big),$$

and we use these terms interchangeably. For simplicity, we omit all the $x$ in the terms and write $f^{(h)} := f^{(h)}(x)$, etc. Write $\dot\sigma^{(h)} = \dot\sigma\Big(f^{(h)}(x)\Big)$. Notice that applying $\sigma(\cdot)$ or Hadamard product with $\dot\sigma^{(h)}$ only decreases norms.

**Lemma B.2** (Probablisitc Moore-Osgood for double sequence). *Let $a_{n,d}$ be a random double sequence in a complete space. Assume for any $\epsilon > 0, \delta \in (0,1)$, there exists $N(\delta) > 0$ and $D(\epsilon) > 0$ such that for all $n > N$ and $d > D$, with probability at least $1-\delta$ we have $|a_{n,d} - a_n| < \epsilon$ (we may refer to this property as uniform convergence with high probability). And for any $d \in \mathbb{N}$ we have $\lim_{n\to\infty} a_{n,d} = a_d$ almost surely, then with high probability:*

$$\lim_{n\to\infty} \lim_{d\to\infty} a_{n,d} = \lim_{d\to\infty} \lim_{n\to\infty} a_{n,d}.$$

*Proof.* We sometimes also write $a_d(n)$ to stress that we consider the sequence as a function of $n$. By assumption, for any $\delta \in (0,1), \epsilon > 0$, there exists $N, D$ such that for all $n > N$, $d, e > D$,

---

[3] Here by "first" we meant the order when you calculate the limits: you first fix $d$ and take the limit of $n$. Not the actual order from left to right.

$|a_d(n) - a_e(n)| < \epsilon$ with probability at least $1 - \delta$. Since here $N$ does not depend on $D$, let $n \to \infty$ we get the following statement holds almost surely:

$$d, e > D \implies |a_d - a_e| < \epsilon \text{ with probability at least } 1 - \delta.$$

This shows that $a_d := \lim_{n \to \infty} a_{n,d}$ is a Cauchy sequence and have a finite limit $\lim_{d \to \infty} a_d = L$.

Now define $a(n) := a_n = \lim_{d \to \infty} a_{n,d}$, for $d > D(\epsilon)$:

$$\left| a(n) - L \right| \leq \underbrace{|a(n) - a_d(n)|}_{A} + \underbrace{|a_d(n) - a_d|}_{B} + \underbrace{|a_d - L|}_{C}.$$

By assumption, pick large enough $n$, we have $A < \epsilon$ with probability at least $1 - \delta$. By the Cauchy sequence argument above, we have $C < \epsilon$ with high probability. Finally since $a_d(n) \to a_d$ pointwise for every $d$, we can choose $n$ large enough such that $B < \epsilon$. This concludes our proof. $\square$

We want to remark that the above Lemma B.2 relies on a more general notion of "conditional almost sure convergence". In particular, we only assume that $|a_{n,d} - a_n| < \epsilon$ almost surely conditioned on an event with probability at least $1 - \delta$:

$$P\left( \lim_{d \to \infty} a_{n,d} = a_n \Big| E \right) = 1, \text{ where } P(E) > 1 - \delta \text{ for all large enough } n.$$

Notice here we are not explicit about how $\delta$ evolves with $n$. When we use this lemma in Theorem 3.5, we have $\delta = o(n)$ which will instead gives us a convergence in probability result. To be complete, we also provide the weaker result and its proof here.

**Lemma B.3** (Another probablisitc Moore-Osgood for double sequence). *Let $a_{n,d}$ be a random double sequence in a complete space. Assume for any $\epsilon > 0$, there exists $D(\epsilon) > 0$ such that for all $d > D$, with probability at least $1 - o(n)$ we have $|a_{n,d} - a_n| < \epsilon$. And for any $d \in \mathbb{N}$ we have $\lim_{n \to \infty} a_{n,d} = a_d$ almost surely, then the following convergence holds in probability:*

$$\lim_{n \to \infty} \lim_{d \to \infty} a_{n,d} = \lim_{d \to \infty} \lim_{n \to \infty} a_{n,d}.$$

*Proof.* We sometimes also write $a_d(n)$ to stress that we consider the sequence as a function of $n$. By assumption, let $n \to \infty$ we get the following statement holds with probability 1:

$$d, e > D \implies |a_d - a_e| < \epsilon.$$

This shows that $a_d := \lim_{n \to \infty} a_{n,d}$ is a Cauchy sequence and have a finite limit $\lim_{d \to \infty} a_d = L$.

Now define $a(n) := a_n = \lim_{d \to \infty} a_{n,d}$, for $d > D(\epsilon)$:

$$\left| a(n) - L \right| \leq \underbrace{|a(n) - a_d(n)|}_{A} + \underbrace{|a_d(n) - a_d|}_{B} + \underbrace{|a_d - L|}_{C}.$$

By assumption, pick large enough $n$, we have $A < \epsilon$ with probability at least $1 - o(n)$. By the Cauchy sequence argument above, we have $C < \epsilon$ with probability 1. Finally since $a_d(n) \to a_d$ pointwise for every $d$, we can choose $n$ large enough such that $B < \epsilon$ with probability at least $1 - o(n)$. Overall this gives

$$P\big( |a(n) - L| > 3\epsilon \big) < o(n),$$

which concludes our proof $\square$

By standard high-dimensional probability (Vershynin, 2019), the following lemma holds:

**Lemma B.4.** *Let $A \in \mathbb{R}^{n \times m}$ be a random matrix whose entries are sampled from i.i.d standard Gaussian distribution, then for $t \geq 0$, with probability at least $1 - e^{-ct^2}$ for a constant $c > 0$, there is:*

$$\|A\|_2 \leq \sqrt{n} + \sqrt{m} + t$$

We are now ready to give the formal proof.

**Theorem 3.5.** *Let $\sigma_W^2 \leq 1/8$, $\Theta_n^{(d)}(x, y) = \sum_{h=1}^{d+1} \left\langle \frac{\partial f(\theta, x)}{\partial \theta^{(h)}}, \frac{\partial f(\theta, y)}{\partial \theta^{(h)}} \right\rangle$ be the empirical NTK with depth $d$ and width $n$. Then $\lim_{n \to \infty} \lim_{d \to \infty} \Theta_n^{(d)} = \lim_{d \to \infty} \lim_{n \to \infty} \Theta_n^{(d)}$ ~~with high probability~~ in probability.*

*Proof of Theorem 3.5.* For any fixed $d$, we write $\Theta^{(d)} = \lim_{n\to\infty} \Theta_n^{(d)}$, notice this is just a finite-depth NTK (possibly with input injection). We condition on the event that $\lim_d \Theta_n^{(d)}$ exists. A sufficient condition for this event to hold with high probability is $\sigma_W^2 < 1/8$. With such $\sigma_W^2$, by Lemma B.4, $\sigma \circ \sqrt{\sigma_W^2/n}W$ has a Lipschitz constant less than 1 with high probability. Recall that $\sigma(x) = \sqrt{2}\max\{0, x\}$ is the normalized ReLU nonlinearity. Conditioned on such event, we have

$$\frac{\partial f(x)}{\partial W^{(h)}}^T \frac{\partial f(x')}{\partial W^{(h)}}$$
$$= g^{(h-1)}(x)^T g^{(h-1)}(x') \cdot p^{(h)}(x)^T p^{(h)}(x')$$
$$\leq \|g^{(h-1)}(x)\|\|g^{(h-1)}(x')\|\|p^{(h)}(x)\|\|p^{(h)}(x')\|$$

WLOG let $g^{(0)} = x \in \mathbb{S}^{d-1}$, and $\|g^{(0)}\| \leq 1$ be our base case. Note that $U^{(h)}x$ is fixed for weight-tied network, let's denote it as $C$, and also overload the notation that $\|C\| = C$. By induction:

$$\left\|g^{(h)}\right\| = \left\|\sigma\left(f^{(h)}\right)\right\| = \left\|\sigma\left(\sqrt{\frac{\sigma_W^2}{n}}W^{(h)}g^{(h-1)} + C\right)\right\|$$
$$\leq \left\|\sqrt{\frac{2\sigma_W^2}{n}}W^{(h)}g^{(h-1)} + C\right\| \leq \sqrt{\frac{2\sigma_W^2}{n}}\left\|W^{(h)}\right\|_{op}\left\|g^{(h-1)}\right\|_2 + \|C\|$$

By Lemma B.4, with probabiliy at least $1 - e^{-\mathcal{O}(t^2)}$, we have $\|W\|_{op} \leq 2\sqrt{n} + t$. This shows that for all $\epsilon > 0$, let $\sigma_W < \frac{1}{2\sqrt{2}+\epsilon}$, with probability at least $1 - e^{-\mathcal{O}(\epsilon^2 n)}$, we have

$$\sqrt{\frac{2\sigma_W^2}{n}}\left\|W^{(h)}\right\|_{op} \triangleq r < 1.$$

Consequently:

$$\|g^{(h)}\| \leq r\|g^{(h-1)}\| + C \leq r^h\|g^{(0)}\| + \sum_{l=1}^{h} Cr^l,$$

which is geometric and converges absolutely as $h \to \infty$. Therefore, there exists a constant $Q > 0$ s.t $\|g^{(h)}\| < Q$ for all $h \in \mathbb{N}$.

By the same spirit, using induction, we have

$$\|p^{(h)}\| \leq \frac{\sqrt{2\sigma_W^2}}{\sqrt{n}}\|W^{(h)}\|_{op}\|p^{(h+1)}\| \leq r\|p^{(h+1)}\| \leq r^{d-h}\|p^{(d+1)}\| = r^{d-h}.$$

Combining the above two derivations, we have

$$\sum_{h=1}^{\infty} \frac{\partial f(x)}{\partial W^{(h)}}^T \frac{\partial f(x')}{\partial W^{(h)}} \leq \sum_{h=1}^{\infty}\left\|\frac{\partial f(x)}{\partial W^{(h)}}\right\|\sum_{h=1}^{\infty}\left\|\frac{\partial f(x')}{\partial W^{(h)}}\right\|$$
$$\leq \left(\sum_{h=1}^{\infty}\|g^{(h-1)}(x)\|\|p^{(h)}(x)\|\right)\left(\sum_{h=1}^{\infty}\|g^{(h-1)}(x')\|\|p^{(h)}(x')\|\right) < \infty.$$

Similar convergence result can be derived for $\frac{df}{dU}$ as well.

Use the terminology introduced in Lemma B.2, $\lim_{d\to\infty} \Theta_n^{(d)} = \lim_{d\to\infty} \Theta^{(d)}(n) = \sum_{h=1}^{\infty} \frac{\partial f(x)}{\partial \theta^{(h)}}^T \frac{\partial f(x')}{\partial \theta^{(h)}}$ converges uniformly in $n$ with high probability.

For a fixed $d$, we know that $\lim_{n\to\infty} \Theta_n^{(d)} = \Theta^{(d)}$ by the tensor program (Yang, 2019). Therefore conditioned on the event that $\sigma \circ \sqrt{\sigma_W^2/n}W$ has a Lipschitz constant less than 1, by Lemma B.2, we can swap the limit and indeed $\lim_{d\to\infty} \lim_{n\to\infty} \Theta_n^{(d)} = \lim_{n\to\infty} \lim_{d\to\infty} \Theta_n^{(d)}$. This shows that the NTK-of-DEQ and the DEQ-of-NTK coincide. □

One should note that it merely requires $\sigma_W^2 < 1$ for the DEQ-of-NTK to converge as in Theorem 3.3, but our above proof requires $\sigma_W^2 < 1/8$ to make sure that the NTK-of-DEQ and DEQ-of-NTK are equivalent. Our current analysis relies heavily on a contraction argument. However, in the actual DEQ setting, it suffice to have $W$ being strongly monotone to guarantee convergence. That is, one only needs the largest eigenvalue of $W$ to be less than 1. This corresponds to have $\sigma_W^2 < 1/2$ (again, this is because we use the normalized ReLU, so there is an extra factor of $\sqrt{2}$) by the semicircular law. We leave the gap to future works.

## C  DETAILS OF SECTION 4

**Theorem 4.1.** *Let $f_n(x)$ be defined as in Equation* (14) *and $\Theta_n^{(d)}$ be the empirical NTK associated with the finite depth approximation of $f_n$ in Equation* (13). *Let $\sigma_W^2 < 1/4$ and $\sigma_W^2 + \sigma_U^2 = 1$. We have*

$$\lim_{d \to \infty} \lim_{n \to \infty} \Theta_n^{(d)} = \lim_{n \to \infty} \lim_{d \to \infty} \Theta_n^{(d)} = \frac{\sigma_v^2 \sigma_U^2 x^T y}{(1 - \sigma_W^2)^2} + \frac{\sigma_v^2 \sigma_U^2 x^T y}{1 - \sigma_W^2}$$

*with high probability.*

*Proof of Theorem 4.1.* Recall that we define $H := \left(I - \sqrt{\frac{\sigma_W^2}{n}} W\right)^{-1}$. This inverse matrix exists with high probability if $\sigma_W^2 < 1/4$, due to a well-known random matrix theory result Lemma B.4. straightforward derivation gives:

$$\lim_{d \to \infty} \left\langle \frac{\partial f_n^{(d)}(x)}{\partial W}, \frac{\partial f_n^{(d)}(y)}{\partial W} \right\rangle$$

$$= \frac{\sigma_U^2 \sigma_v^2}{n} \frac{\sigma_W^2}{n} \left\langle Hv(HUx)^T, Hv(HUx)^T \right\rangle$$

$$= \underbrace{\frac{\sigma_W^2 \sigma_U^2}{n} \left\langle HUx, HUx \right\rangle \frac{\sigma_v^2}{n} \left\langle Hv, Hv \right\rangle}_{A}$$

$$\xrightarrow{p} \underbrace{\sigma_U^2 \sigma_W^2 \sigma_v^2 x^T y \left( \frac{1}{n} \operatorname{tr}\left( H^T H \right) \right)^2}_{B}$$

$$\to \sigma_U^2 \sigma_W^2 \sigma_v^2 x^T y \left( \int \frac{1}{\lambda} d\mu(\lambda) \right)^2 .$$

The first convergence happens with high probability (Arora et al., 2019). Note that $B = \mathbb{E}_{U,v}[A]$. One needs to apply the Gaussian chaos of order 2 lemma (Boucheron et al., 2013) to show the concentration. This was done rigorously down in Arora et al. (2019) Claim E.2. Their proof works for our case as well since we have $\|H^T H\|_2$ bounded independently of $n$ and $d$ with high probability.

The second convergence holds for almost every realization of a sequence of $W$. Recall that $\mu_n$ is the empirical distribution of the eigenvalue of the matrix $\left(I - \sqrt{\frac{\sigma_W^2}{n}} W\right)^T \left(I - \sqrt{\frac{\sigma_W^2}{n}} W\right)$. More precisely, $\mu_n = \frac{1}{n} \sum_{i=1}^n \delta_{\lambda_i}$, $\delta_{\lambda_i}$ is the delta measure at the $i$th eigenvlue value $\lambda_i$. We can rewrite

$$\frac{1}{n} \operatorname{tr}\left( H^T H \right) = \int \frac{1}{\lambda} d\mu_n(\lambda).$$

We will show that $\mu_n \to \mu$ weakly a.s [4]. Then by Portmanteau lemma, we have $\int f d\mu_n \to \int f d\mu$ for every bounded Lipschitz function. Here we have $f = 1/\lambda$ defined when $\lambda$ has non-zero support in $\mu(\lambda)$. Since by Lemma B.4, our assumption $\sigma_W^2 < 1/8$ guarantees $\left\| \sqrt{\frac{\sigma_W^2}{n}} W \right\| < 1$ w.h.p, the support of $\mu(\lambda)$ is bounded away from 0, and $f$ is indeed Lipschitz and bounded on its domain.

---

[4]Note here $\mu_n$ is a random measure

Next, we show that $\int \frac{1}{\lambda} d\mu(\lambda) = \frac{1}{1-\sigma_W^2}$. From Capitaine & Donati-Martin (2016), we learn that the Stieltjes transform $g$ of $\mu$ is a root to the following cubic equation:

$$\text{For } z \in \mathbb{C}^+ : g_\mu(z)^{-1} = \left(1 - \sigma_W^2 g_\mu(z)\right) z - \frac{1}{1 - \sigma_W^2 g_\mu(z)}. \tag{21}$$

Deducing the probability density from $g$ by using the inverse formula of Stieltjes transformation, we have

$$
\begin{aligned}
p(b) &= \lim_{b\to 0^+} \frac{1}{\pi} \text{Im}(g(a+bi)) \\
&= \frac{1}{\pi} \left( \frac{\sqrt{3}\left(3\sigma_W^6 b - \sigma_W^4 b^2 - 3\sigma_W^4 b\right)}{3 \, 2^{2/3} \sigma_W^4 b \left(9\sigma_W^8 b^2 - 2\sigma_W^6 b^3 + 18\sigma_W^6 b^2 + \sqrt{\left(9\sigma_W^8 b^2 - 2\sigma_W^6 b^3 + 18\sigma_W^6 b^2\right)^2 + 4\left(3\sigma_W^6 b - \sigma_W^4 b^2 - 3\sigma_W^4 b\right)^3}\right)^{1/3}} + \right. \\
&\quad \left. \frac{\sqrt{3}\left(9\sigma_W^8 b^2 - 2\sigma_W^6 b^3 + 18\sigma_W^6 b^2 + \sqrt{\left(9\sigma_W^8 b^2 - 2\sigma_W^6 b^3 + 18\sigma_W^6 b^2\right)^2 + 4\left(3\sigma_W^6 b - \sigma_W^4 b^2 - 3\sigma_W^4 b\right)^3}\right)^{1/3}}{6 \sqrt[3]{2} \sigma_W^4 b} \right)
\end{aligned}
$$

Finally we can compute $\int_l^u \frac{1}{\lambda} p(\lambda) d\lambda$. Notice to let $p(\cdot)$ be well defined, we need $9\sigma_W^8 b^2 - 2\sigma_W^6 b^3 + 18\sigma_W^6 b^2 \geq 0$, which amounts to $l = \frac{1}{8}\left(-\sigma_W^4 + 20\sigma_W^2 - \sqrt{\sigma_W^8 + 24\sigma_W^6 + 192\sigma_W^4 + 512a^2} + 8\right)$ and $u = \frac{1}{8}\left(-\sigma_W^4 + 20\sigma_W^2 + \sqrt{\sigma_W^8 + 24\sigma_W^6 + 192\sigma_W^4 + 512a^2} + 8\right)$. This now involves a one-dimensional integral, which an be solved numerically for all values of $\sigma_W$, and shown be be arbitrarily close the desired quantity $1/(1 - \sigma_W^2)$.

Similarly, we can compute that

$$\lim_{d\to\infty} \left\langle \frac{\partial f_n^{(d)}(x)}{\partial U}, \frac{\partial f_n^{(d)}(y)}{\partial U} \right\rangle \xrightarrow{p} \frac{\sigma_v^2 \sigma_U^2 x^T y}{1 - \sigma_W^2}$$

and

$$\lim_{d\to\infty} \left\langle \frac{\partial f_n^{(d)}(x)}{\partial v}, \frac{\partial f_n^{(d)}(y)}{\partial v} \right\rangle \xrightarrow{p} \frac{\sigma_v^2 \sigma_U^2 x^T y}{1 - \sigma_W^2}.$$

Summing the three relevant terms and use the fact that $\sigma_U^2 + \sigma_W^2 = 1$, we get the claimed result. $\square$

## D  DEQ WITH CONVOLUTION LAYERS

In this section we show how to derive the NTKs for convolution DEQs (CDEQ). Although in this paper only the CDEQ with vanilla convolution structure is considered, we remark that our derivation is general enough for other CDEQ structures as well, for instance, CDEQ with global pooling layer. The details of this section can be found in the appendix.

Unlike the fully connection network with input injection, whose intermediate NTK representation is a real number. For convolutional neural networks (CNN), the intermediate NTK representation is a four-way tensor. In the following, we will present the notations, CNN with input injection (CNN-IJ) formulation, the CDEQ-NTK initialization, and our main theorem.

**Notation.**   We adopt the notations from Arora et al. (2019). Let $x, y \in \mathbb{R}^{P\times Q}$ be a pair of inputs, let $q \in \mathbb{Z}_+$ be the filter size (WLOG assume it is odd as well). By convention, we always pad the representation (both the input layer and hidden layer) with 0's. Denote the convolution operation for $i \in [P], j \in [Q]$: $[w * x]_{ij} = \sum_{a=-\frac{q-1}{2}}^{\frac{q-1}{2}} \sum_{b=-\frac{q-1}{2}}^{\frac{q-1}{2}} [w]_{a+\frac{q+1}{2}, b+\frac{q+1}{2}} [x]_{a+i, b+j}$.

Denote

$$\mathcal{D}_{ij,i'j'} = \left\{ (i+a, j+b, i'+a', j'+b') \in [P] \times [Q] \times [P] \times [Q] : -(q-1)/2 \leq a, b, a', b' \leq (q-1)/2 \right\}.$$

Intuitively, $\mathcal{D}_{ij,i'j'}$ is a $q \times q \times q \times q$ set of indices centered at $(ij, i'j')$. For any tensor $T \in \mathbb{R}^{P\times Q\times P\times Q}$, let $[T]_{\mathcal{D}_{ij,i'j'}}$ be the natural sub-tensor and let $\text{Tr}(T) = \sum_{i,j} T_{ij,ij}$.

**Formulation of CNN-IJ.**  Define the CNN-IJ as follows:

- Let the input $x^{(0)} = x \in \mathbb{R}^{P \times Q \times C_0}$, where $C_0$ is the number of input channels, and $C_h$ is the number of channels in layer $h$. Assume WLOG that $C_h = C$ for all $h \in [d]$

- For $h = 1, \ldots, d$, let the inner representation

$$\tilde{x}^{(h)}_{(\beta)} = \sum_{\alpha=1}^{C_{h-1}} \sqrt{\frac{\sigma_W^2}{C_h}} W^{(h)}_{(\alpha),(\beta)} * x^{(h-1)}_{(\alpha)} + \sum_{\alpha=1}^{C_0} \sqrt{\frac{\sigma_U^2}{C_h}} U^{(h)}_{(\alpha),(\beta)} * x^{(0)}_{(\alpha)} \tag{22}$$

$$\left[ x^{(h)}_{(\beta)} \right]_{ij} = \frac{1}{[S]_{ij}} \left[ \sigma \left( \tilde{x}^{(h)}_{(\beta)} \right) \right]_{ij}, \text{ for } i \in [P], j \in [Q] \tag{23}$$

where $W^{(h)}_{(\alpha),(\beta)} \in \mathbb{R}^{q \times q}$ represent the convolution operator from the $\alpha^{th}$ channel in layer $h - 1$ to the $\beta^{th}$ channel in layer $h$. Similarly, $U^{(h)}_{(\alpha),(\beta)} \in \mathbb{R}^{q \times q}$ injects the input in each convolution window. $S \in \mathbb{R}^{P \times Q}$ is a normalization matrix. Let $W, U, S, \sigma_U^2, \sigma_W^2$ be chosen by the CDEQ-NTK initialization described later.

- The final output is defined to be $f_\theta(x) = \sum_{\alpha=1}^{C_d} \left\langle W^{(d+1)}_{(\alpha)}, x^{(d)}_{(\alpha)} \right\rangle$, where $W^{(d+1)}_{(\alpha)} \in \mathbb{R}^{P \times Q}$ is sampled from standard Gaussian distribution.

**CDEQ-NTK initialization.**  Let $1_q \in \mathbb{R}^{q \times q}, X \in \mathbb{R}^{P \times Q}$ be two all-one matrices. Let $\tilde{X} \in \mathbb{R}^{(P+2) \times (Q+2)}$ be the output of zero-padding $X$. We index the rows of $\tilde{X}$ by $\{0, 1, \ldots, P + 1\}$ and columns by $\{0, 1, \ldots, Q + 1\}$. For position $i \in [P], j \in [Q]$, let $\left( [S]_{ij} \right)^2 = [1_q * \tilde{X}]_{ij}$ in Equation (23). Let every entry of every $W, U$ be sampled from $\mathcal{N}(0, 1)$ and $\sigma_W^2 + \sigma_U^2 = 1$.

Using the above-defined notations, we now state the CDEQ-NTK.

**Theorem D.1.** *Let $x, y \in \mathbb{R}^{P \times Q \times C_0}$ be s.t $\|x_{ij}\|_2 = \|y_{ij}\|_2 = 1$ for $i \in [P], j \in [Q]$. Define the following expressions recursively (some $x, y$ are omitted in the notations), for $(i, j, i', j') \in [P] \times [Q] \times [P] \times [Q], h \in [d]$*

$$K^{(0)}_{ij,i'j'}(x, y) = \left[ \sum_{\alpha \in [C_0]} x_{(\alpha)} \otimes y_{(\alpha)} \right]_{ij,i'j'} \tag{24}$$

$$\left[ \Sigma^{(0)}(x, y) \right]_{ij,i'j'} = \frac{1}{[S]_{ij}[S]_{i'j'}} \sum_{\alpha=1}^{C_0} \text{Tr} \left( \left[ K^{(0)}_{(\alpha)}(x, y) \right]_{\mathcal{D}_{ij,i'j'}} \right) \tag{25}$$

$$\mathbb{R}^{2 \times 2} \ni \Lambda^{(h)}_{ij,i'j'}(x, y) = \begin{pmatrix} \left[ \Sigma^{(h-1)}(x, x) \right]_{ij,ij} & \left[ \Sigma^{(h-1)}(x, y) \right]_{ij,i'j'} \\ \left[ \Sigma^{(h-1)}(y, x) \right]_{i'j',ij} & \left[ \Sigma^{(h-1)}(y, y) \right]_{i'j',i'j'} \end{pmatrix} \tag{26}$$

$$\left[ K^{(h)}(x, y) \right]_{ij,i'j'} = \frac{\sigma_W^2}{[S]_{ij} \cdot [S]_{i'j'}} \mathop{\mathbb{E}}_{\substack{(u,v) \\ \sim \mathcal{N}(0, \Lambda^{(h)}_{ij,i'j'})}} [\sigma(u)\sigma(v)] + \frac{\sigma_U^2}{[S]_{ij} \cdot [S]_{i'j'}} [K^{(0)}]_{ij,i'j'} \tag{27}$$

$$\left[ \dot{K}^{(h)}(x, y) \right]_{ij,i'j'} = \frac{\sigma_W^2}{[S]_{ij} \cdot [S]_{i'j'}} \mathop{\mathbb{E}}_{\substack{(u,v) \\ \sim \mathcal{N}(0, \Lambda^{(h)}_{ij,i'j'})}} [\dot{\sigma}(u)\dot{\sigma}(v)] \tag{28}$$

$$\left[ \Sigma^{(h)}(x, y) \right]_{ij,i'j'} = \text{Tr} \left( \left[ K^{(h)}(x, y) \right]_{\mathcal{D}_{ij,i'j'}} \right) \tag{29}$$

*Define the linear operator $\mathcal{L} : \mathbb{R}^{P \times Q \times P \times Q} \to \mathbb{R}^{P \times Q \times P \times Q}$ via $[\mathcal{L}(M)]_{ij,i'j'} = \text{Tr} \left( [M]_{\mathcal{D}_{ij,i'j'}} \right)$.*

*Then the CDEQ-NTK can be found solving the following linear system:*

$$\Theta^*(x,y) = \dot{K}^*(x,y) \odot \mathcal{L}(\Theta^*(x,y)) + K^*(x,y), \tag{30}$$

*where $K^*(x,y) = \lim_{d\to\infty} K^{(L)}(x,y)$, $\dot{K}^*(x,y) = \lim_{d\to\infty} \dot{K}^{(d)}(x,y)$. The limit exists if $\sigma_W^2 < 1$. The actual NTK entry is calculated by $\mathrm{Tr}(\Theta^*(x,y))$.*

Theorem D.1 highlights that the convergence of CDEQ-NTK depends solely on the CDEQ-NTK initialization. The crucial factor here is the normalization tensor $S$, which guarantees the variance of each term is always 1 across the propogation. This idea mimics that of the DEQ-NTK initialization. Our theorem shows that CDEQ-NTK can also be computed by solving fixed point equations.

We first explain the choice of $S$ in the CDEQ-NTK initialization. In the original CNTK paper (Arora et al., 2019), the normalization is simply $1/q^2$. However, due to the zero-padding, $1/q^2$ does not normalize all $\left[\Sigma^{(h)}(x,x)\right]_{ij,i'j'}$ as expected: only the variances that are away from the corners are normalized to 1, but the ones near the corner are not. $[S]_{ij}$ is simply the number of non-zero entries in $\left[\tilde{X}\right]_{\mathcal{D}_{ij,ij}}$.

Now we give the proof to Theorem D.1.

*Proof of Theorem D.1.* Similar to the proof of Theorem 3.1, we can split the CDEQ-NTK in two terms:

$$\Theta^{(L)}(x,y) = \mathbb{E}_\theta\left[\left\langle \frac{\partial f(\theta,x)}{\partial\theta}, \frac{\partial f(\theta,y)}{\partial\theta}\right\rangle\right]$$

$$=\underbrace{\mathbb{E}_\theta\left[\left\langle \frac{\partial f(\theta,x)}{\partial W}, \frac{\partial f(\theta,y)}{\partial W}\right\rangle\right]}_{\text{①}} + \underbrace{\mathbb{E}_\theta\left[\left\langle \frac{\partial f(\theta,x)}{\partial U}, \frac{\partial f(\theta,y)}{\partial U}\right\rangle\right]}_{\text{②}}.$$

Omit the input symbols $x, y$, let

$$\left[\widehat{K}^{(h)}\right]_{ij,i'j'} = \frac{\sigma_W^2}{[S]_{ij} \cdot [S]_{i'j'}} \mathop{\mathbb{E}}_{(u,v)\sim\mathcal{N}(0,\Lambda_{ij,i'j'}^{(h)})} [\sigma(u)\sigma(v)].$$

As shown in Arora et al. (2019), we have

$$\left\langle \frac{\partial f_\theta(x)}{\partial W^{(h)}}, \frac{\partial f_\theta(,y)}{\partial W^{(h)}}\right\rangle \to \mathrm{Tr}\left(\dot{K}^{(d)} \odot \mathcal{L}\left(\dot{K}^{(d-1)} \odot \mathcal{L}\left(\cdots \dot{K}^{(h)} \odot \mathcal{L}\left(\widehat{K}^{h-1}\right)\cdots\right)\right)\right)$$

Write $f = f_\theta(x)$ and $\tilde{f} = f_\theta(y)$. Following the same step, by chain rule, we have

$$\left\langle \frac{\partial f}{\partial U^{(h)}}, \frac{\partial \tilde{f}}{\partial U^{(h)}}\right\rangle \to \mathrm{Tr}\left(\dot{K}^{(d)} \odot \mathcal{L}\left(\dot{K}^{(d-1)} \odot \mathcal{L}\left(\cdots \dot{K}^{(h)} \odot \mathcal{L}\left(K^{(0)}\right)\cdots\right)\right)\right)$$

Rewrite the above two equations in recursive form, we can calculate the $L$-depth iteration of CDEQ-NTK by:

- For the first layer $\Theta^{(0)}(x,y) = \Sigma^{(0)}(x,y)$.

- For $h = 1, \ldots, d-1$, let

$$\left[\Theta^{(h)}(x,y)\right]_{ij,i'j'} = \mathrm{Tr}\left(\left[\dot{K}^{(h)}(x,y) \odot \Theta^{(h-1)}(x,y) + K^{(h)}(x,y)\right]_{\mathcal{D}_{ij,i'j'}}\right) \tag{31}$$

- For $h = d$, let

$$\Theta^{(L)}(x,y) = \dot{K}^{(d)}(x,y) \odot \Theta^{(d-1)}(x,y) + K^{(h)}(x,y) \tag{32}$$

- The final kernel value is $\mathrm{Tr}(\Theta^{(d)}(x, y))$.

Using Equation (31) and Equation (32), we can find the following recursive relation:

$$\Theta^{(d+1)}(x, y) = \dot{K}^{(d+1)}(x, y) \odot \mathcal{L}\left(\Theta^{(d)}(x, y)\right) + K^{(h+1)}(x, y) \tag{33}$$

The rest of the proof is stated in the main text. For readers' convenience we include them here again.

At this point, we need to show that $K^*(x, y) \triangleq \lim_{d \to \infty} K^{(d)}(x, y)$ and $\dot{K}^*(x, y) \triangleq \lim_{d \to \infty} \dot{K}^{(d)}(x, y)$ exist. Let us first agree that for all $h \in [d]$, $(ij, i'j') \in [P] \times [Q] \times [P] \times [Q]$, the diagonal entries of $\Lambda_{ij,i'j'}^{(h)}$ are all ones. Indeed, these diagonal entries are 1's at $h = 0$ by initialization. Note that iterating Equations (26) to (29) to solve for $[\Sigma^{(h)}(x, y)]_{ij,i'j'}$ is equivalent to iterating $f : \mathbb{R}^{P \times Q \times P \times Q} \to \mathbb{R}^{P \times Q \times P \times Q}$:

$$P^{(h+1)} = f(P^{(h)}) \triangleq \mathcal{L}\left(\frac{1}{[S]_{ij}[S]_{i'j'}} R_\sigma(P^{(h)})\right), P^{(0)} = K^{(0)} \tag{34}$$

where

$$R_\sigma(P_{ij,i'j'}^{(h)}) \triangleq \sigma_W^2 \left(\frac{\sqrt{1 - \left(P_{ij,i'j'}^{(h)}\right)^2} + \left(\pi - \cos^{-1}\left(P_{ij,i'j'}^{(h)}\right)\right) P_{ij,i'j'}^{(h)}}{\pi}\right) + \sigma_U^2 K_{ij,i'j'}^{(0)} \tag{35}$$

is applied to $P^{(h)}$ entrywise.

Due to CDEQ-NTK initialization, if $P_{ij,ij}^{(0)} = 1$ for $i \in [P], j \in [Q]$, then $P_{ij,ij}^{(h)} = 1$ for all iterations $h$. This is true by the definition of $S$.

Now if we can show $f$ is a contraction, then $\Sigma^*(x, y) \triangleq \lim_{h \to \infty} \Sigma^{(h)}(x, y)$ exists, hence $K^*$ and $\dot{K}^*$ also exist. We should keep the readers aware that $f : \mathbb{R}^{P \times Q \times P \times Q} \to \mathbb{R}^{P \times Q \times P \times Q}$, so we should be careful with the metric spaces. We want every entry of $\Sigma^{(h)}(x, y)$ to converge, since this tensor has finitely many entries, this is equivalent to say its $\ell^\infty$ norm (imagine flattenning this tensor into a vector) converges. So we can equip the domain an co-domain of $f$ with $\ell^\infty$ norm (though these are finite-dimensional spaces so we can really equip them with any norm, but picking $\ell^\infty$ norm makes the proof easy).

Now we have $f = \mathcal{L} \circ \frac{1}{[S]_{ij}[S]_{i'j'}} R_\sigma : \ell^\infty \to \ell^\infty$. If we flatten the four-way tensor $P^{(h)}$ into a vector, then $\mathcal{L}$ can be represented by a $(P \times Q \times P \times Q) \times (P \times Q \times P \times Q)$ dimensional matrix, whose $(kl, k'l')$-th entry in the $(ij, i'j')$-th row is 1 if $(kl, k'l') \in \mathcal{D}_{ij,i'j'}$, and 0 otherwise. In other words, the $\ell^1$ norm of the $(ij, i'j')$-th row represents the number of non-zero entries in $\mathcal{D}_{ij,i'j'}$, but by the CDEQ-NTK initialization, the row $\ell^1$ norm divided by $[S]_{ij} \cdot [S]_{i'j'}$ is at most 1! Using the fact that $\|\mathcal{L}\|_{\ell^\infty \to \ell^\infty}$ is the maximum $\ell^1$ norm of the row, and the fact $R_\sigma$ is a contraction (proven in Theorem 3.3), we conclude that $f$ is indeed a contraction.

With the same spirit, we can also show that Equation (32) is a contraction if $\sigma_W^2 < 1$, hence Equation (30) is indeed the unique fixed point. This finishes the proof. $\qquad\square$

### D.1 COMPUTATION OF CDEQ-NTK

One may wish to directly compute a fixed point (or more precisely, a fixed tensor) of $\Theta^{(d)} \in \mathbb{R}^{P \times Q \times P \times Q}$ like Equation (10). However, due to the linear operator $\mathcal{L}$ (which is just the ensemble of the trace operator in Equation (29)), the entries depend on each other. Hence the system involves a $(P \times Q \times P \times Q) \times (P \times Q \times P \times Q)$-dimensional matrix that represents $\mathcal{L}$. Even if we exploit the fact that only entries on the same "diagonal" depend on each other, $\mathcal{L}$ is at least $P \times Q \times P \times Q$, which is $32^4$ for CIFAR-10 data.

Moreover, this system is nonlinear. Therefore we cannot compute the fixed point $\Sigma^*$ by root-finding efficiently. Instead, we approximate it using finite depth iterations, and we observe that in experiments they typically converge to $10^{-6}$ accuracy in $\ell^\infty$ within 15 iterations.

Table 2: Performance of CDEQ-NTK on CIFAR-10 dataset

| Method | Parameters | Acc. |
|---|---|---|
| CDEQ-NTK with 2000 training data | $\sigma_W^2 = 0.65, \sigma_U^2 = 0.35$ | 37.49% |
| CNTK with 2000 training data | Depth = 6 | 43.43% |
| CNTK with 2000 training data | Depth = 21 | 42.53% |

Table 3: Performance of DEQ-NTK on CIFAR-10 dataset, see Lee et al. (2020) for NTK with ZCA regularization..

| Method | Parameters | Acc. |
|---|---|---|
| DEQ-NTK | $\sigma_W^2 = 0.25, \sigma_U^2 = 0.25, \sigma_b^2 = 0.5$ | 59.08% |
| DEQ-NTK | $\sigma_W^2 = 0.6, \sigma_U^2 = 0.4, \sigma_b^2 = 0$ | **59.77%** |
| DEQ-NTK | $\sigma_W^2 = 0.8, \sigma_U^2 = 0.2, \sigma_b^2 = 0$ | 59.43% |
| NTK with ZCA regularization | $\sigma_W^2 = 2, \sigma_b^2 = 0.01$ | 59.7% |

We test CDEQ-NTK accuracy on CIFAR-10 dataset with just 2000 training data. The result is shown in Table 2.

Table 4: Performance of DEQ-NTK on MNIST dataset, compared to neural ODE (Chen et al., 2018b) and monotone operator DEQ, see these results from Winston & Kolter (2020).

| MNIST | | |
|---|---|---|
| **Method** | **Model size** | **Acc.** |
| DEQ-NTK | | **98.6%** |
| Neural ODE | 84K | 98.2% |
| MON DEQ | 84K | 98.2% |

