# OpenReview forum: "On the Neural Tangent Kernel of Equilibrium Models"
_ICLR.cc/2023/Conference — Submitted to ICLR 2023_

### Official Review · Reviewer_TcTa · 2022-10-24

**Confidence:** 3
**Correctness:** 2
**Technical Novelty And Significance:** 2
**Empirical Novelty And Significance:** 1
**Recommendation:** 3

**Clarity, Quality, Novelty And Reproducibility:**

**Clarity:**
Unfortunately, there are issues around clarity. See detailed comments above.

**Novelty:**
As mentioned above, the key ingredient of this paper this paper is writing the first equation of section 3 as a network without weight-tying, which appears to be an existing result of Yang. Once this can be written, standard techniques apply to analyse the NTK. The reader is not given any assistance in understanding if or how the result of Yang can be applied.

**Reproducibility:**
There are some minor issues in the experiments section.

**Strength And Weaknesses:**

**Strengths:**
- Studying the NTK of a DEQ is an interesting mathematical problem, and will help connect the two communities of implicit neural networks and infinite width neural networks.

**Weaknesses**
- A very critical part of the proof and intuition for this paper is that the NTK of a weigh-tied network is the same as the NTK of a corresponding network with independently sampled weights in each layer. This appears to be an existing result of Yang. However, as this a very central part of the paper (first equations in section 3), it needs to be thoroughly unpacked and explained in the main text. Currently, the way the analysis is presented is backwards (the unweight-tied version is introduced first), and remark 3.2 gives no hint of the machinery behind the statement. The proof of theorem 3.1 in the appendix seems to also not be helpful in this issue. As such, at the moment I can only see that this paper discusses infinitely deep networks with independently sampled weights in each layer, not DEQs. What are the conditions and/or assumptions required for the equivalence of weight tied and not weight tied networks?
- Theorem 3.1 and 3.3 seem to give no conditions that ensure that that the fixed point is unique, or even that exists. In contrast, the condition in Theorem 3.5 that the standard deviation is less than 1/8 seems to be a contraction condition that ensures that a unique fixed point exists. I do not think Theorem 3.1 and 3.3 can be correct in general without some assumption ensuring that a fixed point is unique and/or is unique. But at the moment, the text preceding the theorem is not precise enough to even allow the theorem to be expressed in terms of such conditions or assumptions. Can you add the required assumptions and conditions into the relevant theorems?
- There are some further issues around clarity (detailed below).
- As far as I can tell (please correct me if I am wrong), the paper makes no attempt to argue that the derived NTK-DEQ is a practically useful model, only that it is a mathematical curiosity. Given the toy evaluation, it is unlikely that such a model will be useful in the broader context of machine learning.


**Mathematical ambiguities and correctness:**
- Section 3. In this equation block, this infinite depth limit does not always converge to the unique fixed point. Since this seems to be the main mathematical section of the paper, it is necessary to have some more precision in this section.
- First equation in section 3. This is not a weight-tied network, so the text "Define the ... DEQ as the following:" is not appropriate. You mention later that DEQs require weight-tied networks and that the limit of this un weight-tied network will be the same as the weight-tied. If this is the case, why not introduce the DEQ with weight tying and then prove that this limit is the same as with weight-tying? As is, the sentence before the equation is not accurate.
- In order to invoke the result of Yang, you mention that "The neural architecture needs to satisfy a gradient independent assumption." Can you write down what this assumption is?
- Proof of Theorem 3.3. For equations (17) and (18), it might be more appropriate to cite Cho and Saul than Daniely, since Cho and Saul were first.
- Theorem 3.5. Please give a definition of "with high probability".

**Experiments:**
- Section 5.2. What is the model? I assume some kind of kernel method? Kernel logistic regression? SVM? In the text it says "After obtaining the NTK matrix, we apply kernel regressions", but these problems look like classification problems?
- "Notice these root finding problems are one-dimensional, hence can be quickily solved." Which problems are "these", and why are they one-dimensional? It is not obvious to me as currently written. For a dataset of size $n$, each data point will appear in the kernel matrix $n$ times (granted, it is a symmetric matrix). There is a typo in quickly.
- Section 5.1. Are these $W$ trained in an NTK regime? (Non-stochastic) gradient descent? What is the data and task that the network was trained on?
- Figure 5. Which nonlinearities are used in the figure on the right?

**Minor:**
- First sentence of introduction. There are also optimisation-based layers (Deep Declarative Networks: A New Hope), which implicitly define the output of a layer as the solution to a (possibly constrained) optimisation problem. Under sufficient regularity, this is equivalent to finding a root of the gradient. But not always. These are also mentioned in the implicit layer NeurIPS tutorial.
- "Bai et al. (2019) proposed the DEQ model, which is equivalent to running an infinite-depth FCNN-IJ, but updated in a more clever way". This is not precise. A DEQ model *can* be equivalent to an infinite-depth FCNN-IJ. For example, if the mapping involved in each iteration is a contraction, the infinite-depth recursion converges to the unique fixed point. But a DEQ model is not necessarily *always* equivalent to an infinite-depth FCNN-IJ. You mention existence and uniqueness at the end of the paragraph, but this should come first.
- Two periods on "bias terms.." in section 3.




**Summary Of The Paper:**

Two interesting machine learning models that are simultaneously of theoretical and practical interest are Deep Equilibrium Models (DEQs) and well-behaved infinite-width neural networks trained using gradient flow (Neural tangent kernel, NTK). The first is theoretically interesting, because it can *sometimes* be used to understand very deep weight-tied networks, and the second is interesting because it can *sometimes* be used to understand finite-width neural networks trained using gradient descent. This paper combines elements from DEQs and NTKs, showing how the NTK of a DEQ can be computed in a certain regime.

**Summary Of The Review:**

This paper analyses DEQs in the context of the NTK. My biggest concern is that as currently written, it is not clear that the weight-tied structure of DEQs is faithfully analysed (see section 3 first equation and remark 3.2). If this structure is indeed not faithfully analysed, the analysis reduces to a trivial known result. The writing in section 3 could generally be improved (the lack of precision of the assumptions and conditions in section 3 arguably makes them incorrect), as could the details of the experiments. The experiments do not seem to argue that the derived model is practically useful (which would not be an issue with the paper in itself, if its other merits could stand on their own).

---

> ### Author Response · Authors · 2022-11-17
> **Thanks for your review**
>
> 1. untied vs tied weights: We have a more detailed proof section in the revision appendix now, please see appendix A. Tensor/nestor program is quite notation heavy and we try to avoid this. The equivalence has been demonstrated previously so this is why we choose this way. The key takeaway from the tensor program is that as long as your last layer is sampled independently from everything else, you can calculate the NTK as if the internal weights are sampled independently.
> 2. uniqueness of fixed point. In theorem 3.1, we are only talking about the "DEQ-of-NTK". For this fixed point to be unique, we need $\sigma_W^2<1$. The existence of the fixed point is justified in lemma B.1 in the appendix. The uniqueness comes from the way we prove theorem 3.3. The proof also relies on a contraction argument, so by Banach fixed point theorem, the fixed point is unique. Theorem 3.5 is talking about "NTK-of-DEQ", and when $\sigma_W^2<1/8$, we can say the "NTK-of-DEQ" equals "DEQ-of-NTK" (put differently, if $\sigma_W^2\geq 1/8$, these two notions may or may not be equivalent). There is no contradiction. Although both theorem 3.3 and theorem 3.5 use contractions, in theorem 3.3 we take width to the infinity first, and in theorem 3.5 we take depth to the infinity first. This is why they lead to different constraints on $\sigma_W^2$.
> 3. "with high probability": should be "in probability" in theorem 3.5.
> 4. experiment model: it's a kernel method, where the kernel is of size $n\times n$, n is the number of training data. Each entry in the kernel is computed as in theorem 3.3. The classification is solved using linear regression. Say for a $10$-class classification, the target vector is one-hot in $R^{10}$, and we change the $0$s to $-0.1$s, and $1$ to $0.9$. This has been done previously in [1] and [2]. $\textbf{EDIT}:$ We realized that some questions may not be fully addressed before this edit. In section 5.1, the $W$ is "trained" in the NTK regime. However, you don't need to actually train anything since the NTK matrix doesn't change. You can just use the NTK matrix (computed by theorem 3.3) to perform kernel regression, which captures the behavior of a wide NN in function space.
> 5. one-dimensional: each entry of the kernel matrix is one-dimensional, although there are in total $n^2$ entries. Since each entry is 1d and there is no dependency among entries, we can efficiently parallelize them.
> 6. nonlinearity: all nonlinearities in this paper is normalized relu: $\sqrt{2}\max(x, 0)$
>
> $\textbf{EDIT}$: we apologize that in the proof of theorem 3.3 Cho and Saul was not properly addressed in the current revision (unfortunately the revision deadline has passed), we will make sure citation is fixed properly in the future version.
>
>
> We have also modified other wording/citation issues.
>
> [1]: On Exact Computation with an Infinitely Wide Neural Net
> [2]: Bayesian deep convolutional networks with many channels are gaussian processes.

---

### Official Review · Reviewer_fAKE · 2022-10-26

**Confidence:** 4
**Correctness:** 3
**Technical Novelty And Significance:** 3
**Empirical Novelty And Significance:** 3
**Recommendation:** 5

**Clarity, Quality, Novelty And Reproducibility:**

Clarity: The paper writing is clear in general while it could be further improved.

Quality & Novelty: The proof builds upon exisiting method, the real novelty should be made clear further.

Reproducibility: Given the dense mathematics and some minor issues, it is not very easy to reproduce the method.

**Strength And Weaknesses:**

Strength:

+ The paper showed that unlike the infinite depth limit of NTK to FCNN, the DEQ-of-NTK does not converge to a degenerate kernel. This non-trivial kernel can be computed efficiently using root-finding.

+ The NTK-of-DEQ coincides with the DEQ-of-NTK under mild conditions. The paper showed numerically that reasonably large networks converge to roughly the same quantities as predicted by theory.

+ The paper showed the NTK-of-DEQ matches the performances of other NTKs on real-world datasets.

Weaknesses:

- Only simulation experiments are conducted to demonstrate the performance of the NTK-of-DEQ.

- The paper is presented in a rather dense mannar. It is better to give a high-level overview of the proof pipeline.

**Summary Of The Paper:**

The paper showed that contrarily a DEQ model enjoys a deterministic NTK despite its width and depth going to infinity at the same time under mild conditions. The deterministic NTK can be found efficiently via root-finding.

**Summary Of The Review:**

The paper showed that contrarily a DEQ model enjoys a deterministic NTK despite its width and depth going to infinity at the same time under mild conditions. The deterministic NTK can be found efficiently via root-finding. The main contributions of the paper lie in the theoretical aspects and simulations have demonstrated the performance. Some of the key proof builds upon existing work, the real new contributions should be made clear.

---

> ### Author Response · Authors · 2022-11-17
> **Thanks for your review**
>
> We used to have experiment results on real datasets but this often seems to digress audiences from the the point we want to make. We add them to the appendix in the revision. See appendix D.1 table 3 and 4.

---

### Official Review · Reviewer_4dFG · 2022-10-27

**Confidence:** 2
**Correctness:** 4
**Technical Novelty And Significance:** 3
**Empirical Novelty And Significance:** Not applicable
**Recommendation:** 5

**Clarity, Quality, Novelty And Reproducibility:**

The main structure and logistics of this paper are quite clear and easy to follow. But some notations are not defined unambiguously enough.
The major technical innovative results of this paper is to prove a nested limits could be exchanged in Theorem 3.5. With this holding true, one can conclude the existence of deterministic NTK-of-DEQ.

**Strength And Weaknesses:**

Strength
- This paper shows that unlike the infinite depth limiting case of NTK to FCNN, the DEQ-of-NTK does not converge to a degenerate kernel, and the non-trivial kernel can be computed efficiently using root-finding.
- The forms of the final main theorems are compact, intuitive and easy to understand.

Weakness and Questions
- What is the significance or any application of the conclusion that the DEQ-of-NTK and NTK-of-DEQ are deterministic? I understand this is a theoretical work, but I think it is better to mention the usage or theoretical significance for the existence of a deterministic NTK-of-DEQ as I'm not familiar with the idea of DEQ.
- What is the accurate meaning of "high probability" in Theorem 3.5. Is it somehow related to the distance between $\sigma_w^2$ and 1/8?
- I'm a bit confused about the definition of $\sigma_W$ and the initialization process of $W,U,b,v$. The author mentioned in the beginning of Section 3.1 that pick $\sigma_W,\sigma_U,\sigma_b$ arbitrarily . But it seems there is no constraint for $\sigma_W$ in the beginning of Section 3.1. What is the difference between this one and that defined in Theorem 3.1.
- What will happen if $\sigma_W^2>1/8$ ? Will the NTK-of-DEQ definitely diverges or there be some possibility that both NTK-of-DEQ and DEQ-of-NTK converges but at different limits?

**Summary Of The Paper:**

This work studies the NTK of the dee equilibrium model.  Contrast to the NTK of FCNN which can be stochastic if its width and depth both tend to infinity simultaneously, a DEQ model will have a deterministic NTK in this case under some mild conditions. Also this deterministic NTK can be found efficiently via root-finding.

**Summary Of The Review:**

I'm not familiar with DEQ model, but in my perspective this paper made some theoretical, especially technical contribution for proving the existence of NTK-to-DEQ by proving an exchangeable nested limit. However I'm not sure what further theoretical or application results can be induced from this conclusion.

---

> ### Author Response · Authors · 2022-11-17
> **Thanks for your review**
>
> 1. It is mainly a mathematical curiosity to derive the NTK of DEQ, and it's quite surprising that you can switch limits. One possible way to get out of this work is that one should probably initialize DEQ weights with a small scaling to make the training more stable.
> 2. It should be "in probability". See the newly added remark 3.6. In particular $\Theta_n$ is a random sequence and it converges in probability.
> 3. For the result in theorem 3.1 you don't need constraints on the $\sigma$s. For the result in section 3.2, you do need their sum to be $1$. (Maybe I'm not interpreting your question correctly, but "arbitrarily" means no constraint. Do you mean section 3.2? )
> 4. Our results don't specify anything for $\sigma_W^2\geq 1/8$. Although we conjecture the tighter bound is $\sigma_W^2<1/2$. At least for the linear DEQ our current bound seems to be loose (see figure 4(b)). This is left for future work.

---

### Author Response · Authors · 2022-11-17
**To all reviewers**

We thank every reviewers' comments and suggestions. Here is an overview to the rebuttal revision. All major modifications are written in blue.
1. We provide a new appendix A that gives a background on Nestor program from [1]. This justifies why we can work with weight-tied networks.
2. Theorem 3.5 should be "in probability" instead of "with high probability". An explanation is given in remark 3.6, and further discussion can be found in appendix B.1.

[1]: Tensor Programs II: Neural Tangent Kernel for Any Architecture

---

### Decision · Program_Chairs · 2023-01-20

**Decision:**

Reject

**Justification For Why Not Higher Score:**

Technical issues in the main results are the critical reason. Moreover, the numerical evidence should have a clearer demonstration of the usefulness of the theory.

**Justification For Why Not Lower Score:**

N/A

**Metareview: Summary, Strengths And Weaknesses:**

The work derives the NTK for deep equilibrium  (DEQ) models in a certain regime of scaling. The key trick is the exchange of the limit for the depth and width and it is not clear if this is already not possible. For instance, I would have expected a comparison with the paper "Neural Tangent Kernel Beyond the Infinite-Width Limit: Effects of Depth and Initialization" ICML2022.

There are some proof issues with Theorems 3.1 and 3.3. that are not fixed during the discussion period. There are also several issues in clarity, in particular the lack of precision in the assumptions and conditions in Section 3. The numerical evidence also does not support the usefulness of the theory.